# Natural Taxanes: From Plant Composition to Human Pharmacology and Toxicity

**DOI:** 10.3390/ijms232415619

**Published:** 2022-12-09

**Authors:** Ľuboš Nižnanský, Denisa Osinová, Roman Kuruc, Alexandra Hengerics Szabó, Andrea Szórádová, Marián Masár, Žofia Nižnanská

**Affiliations:** 1Department of Forensic Medicine and Toxicology, Health Care Surveillance Authority, Antolská 11, 85107 Bratislava, Slovakia; 2Institute of Forensic Medicine, Faculty of Medicine, Comenius University in Bratislava, Sasinková 4, 81108 Bratislava, Slovakia; 3Department of Anaesthesiology and Intensive Care, Jessenius Faculty of Medicine in Martin, Comenius University in Bratislava, Malá Hora 4a, 03601 Martin, Slovakia; 4Department of Chemistry, Faculty of Education, J. Selye University, Bratislavská cesta 3322, 94501 Komárno, Slovakia; 5Department of Analytical Chemistry, Faculty of Natural Sciences, Comenius University in Bratislava, Mlynská Dolina, Ilkovičova 6, 84215 Bratislava, Slovakia

**Keywords:** natural taxanes, paclitaxel, plant composition, pharmacokinetics, toxicity, intoxication, *Taxus baccata*, death

## Abstract

Biologically active taxanes, present in small- to medium-sized evergreen conifers of various *Taxus* species, are widely used for their antioxidant, antimicrobial and anti-inflammatory effects, but mostly for their antitumour effects used in the treatment of solid tumours of the breast, ovary, lung, bladder, prostate, oesophagus and melanoma. More of the substances found in *Taxus* plant extracts have medical potential. Therefore, at the beginning of this review, we describe the methods of isolation, identification and determination of taxanes in different plant parts. One of the most important taxanes is paclitaxel, for which we summarize the pharmacokinetic parameters of its different formulations. We also describe toxicological risks during clinical therapy such as hypersensitivity, neurotoxicity, gastrointestinal, cardiovascular, haematological, skin and renal toxicity and toxicity to the respiratory system. Since the effect of the drug-form PTX is enhanced by various *Taxus* spp. extracts, we summarize published clinical intoxications and all fatal poisonings for the *Taxus baccata* plant. This showed that, despite their significant use in anticancer treatment, attention should also be focused on the risk of fatal intoxication due to ingestion of extracts from these plants, which are commonly found in our surroundings.

## 1. Introduction

Over millions of years, plants have evolved and adapted to withstand bacteria, insects, fungi, animals and humans, and extreme conditions, and to produce unique, structurally diverse secondary metabolites. Plants have been documented for their medicinal uses for thousands of years, and the use of natural products as medicinals is described throughout history in the form of traditional medicines, remedies, decoctions and oils. However, many of these bioactive natural products as well as their medicinal effects have not yet been described.

Historically, the *Taxus* Gymnosperms are an ancient plant group that originated in the Permian Era, 200–300 million years ago. Although they are outnumbered by angiosperms in terms of species richness, gymnosperms, especially conifers, dominate the world’s forest types and occur in vast temperate landscapes of both hemispheres [1]. *Taxaceae* are a unique family among gymnosperms. The species name *Taxus* comes from the poisonous taxanes found in parts of the tree. Some botanists did not consider yew a true conifer because it does not bear seeds in a cone. However, a proper consideration of its evolutionary relationships now places the Taxaceae family firmly among the conifers [1]. Eight genera of the *Taxus* species are typically recognized. *Taxus baccata* (European or English yew), *Taxus brevifolia* (Pacific or western yew), *Taxus canadensis* (Canadian yew), *Taxus chinensis* (Chinese yew), *Taxus cuspidata* (Japanese yew), *Taxus floridana* (Florida yew), *Taxus globosa* (Mexican yew) and *Taxus wallichiana* (Himalayan yew) [2]. *Taxus baccata* or European yew is widespread in the temperate zones of the northern hemisphere. It is a small- to medium-sized evergreen tree that has historically been used for weapons and medicine. Apart from the fleshy skin of the fruit, it is poisonous. Zhou et al. analysed a total of 2246 substances of primary and secondary metabolism, of which in *T. media*, *T. cuspidata* and *T. maieri*, most substances were part of the “diterpenoid biosynthesis” pathway, which produces paclitaxel (PTX). The components of individual taxanes differ between individual plants and in individual plant parts, and their amount also depends on the actual season [3]. PTX concentration in *T. cuspidata, T. media, and T. mairei* is 1.67, 1.22, and 0.66 mg/g, respectively. Paclitaxel (Taxol A) was isolated from *Taxus* plants and has been widely used since 1984 in the treatment of various cancers (mainly tumours of the breast, ovaries, pancreas or stomach) [4], while in order to improve the effect, new ways of administering this drug are constantly being developed [5,6,7,8,9]. The reason is mainly the achievement of a better biological effect in the exact target place in different types of tissues [10,11] with the least possible number of side effects and at the same time bypassing various resistant mechanisms of tumour cells (overexpression of multidrug resistance (MDR) efflux transporters or by microtubule alterations) [8,12]. Therefore, it is good to know both how paclitaxel works when administered by itself and how this drug behaves when it is part of the plant in which it occurs naturally. For example, in an animal study, 10-Deacetylbaccatine III (10-DAB III, also a taxane with antitumor effect) reached higher concentrations in the blood, lungs, kidney, brain, liver, heart, and spleen throughout the time range of concentration measurement when *T. chinensis* extract was administered compared to the administration of pure 10-DAB III substance [11]. In various plants as well as *Taxus* spp., different mechanisms have been observed where the components of the same plant can have a synergistic effect of biologically active substances, which can ultimately have a positive or negative effect [13,14,15]. Recently, one of the improvements in the effect of PTX is its combination with the *Taxus* extract itself. In an animal study, it was observed that orally administered PTX together with *T. yunnanensis* extract increased its pharmacokinetic parameters (area under curve (AUC) and maximum serum concentration (C_max_)) due to the inhibition of PTX efflux in the gastrointestinal tract. In addition, it was later observed that this extract potentiates PTX-induced cytotoxicity, i.e., exerts an anti-cancer effect through oral administration [16]. Similarly, it was also observed in the combination of PTX with other extracts from *Taxus* spp. (e.g., *T. chinensis*), where the distribution of PTX into the tissues also increased [17]. Originally, paclitaxel was commercially isolated from the bark of *Taxus baccata*, the European yew tree, which contains the greatest amount of this compound among all kinds of yew [15]. In addition to PTX, natural taxanes extracted from *Taxus* spp. (mainly *T. baccata*) contain a wide range of alkaloids, the most famous of which are: taxine A, taxine B and paclitaxel (taxol A), as well as other alkaloids milosine, ephedrine, taxicatin glycoside and their derivatives and natural dyes from the flavonoid group [18]. The above-mentioned molecules could have a mutually supportive and synergistic effect [19]. 

When using PTX or extracts from *Taxus* spp. or their combination, especially with *T. baccata*, however, one must be extremely careful since its use is associated with high cardio-toxicity described in several clinical case reports [20]. Moreover, its use has been linked to several deaths [21,22].

This comprehensive review offers a strong basis for choosing appropriate methods for the isolation, identification and quantification of new or already known substances found in *Taxus* spp. One of these substances is paclitaxel, belonging to the most important chemotherapy drugs, for which we describe the basic mechanism of action, pharmacokinetics and pharmacodynamics in humans. The importance of this article can also be seen in the last sections, where the adverse effects of PTX treatment on organ systems, mechanisms of toxicity, and intoxication are summarized not only in the case of PTX, but also in the case of the use of the plant itself, often ending in death. We also point out that in addition to the observed unwanted toxic effects during treatment, their correlation with actual blood levels of taxane would be useful, especially in cases of patients non-responding to chemotherapy. It is similar to clinical cases of poisoning, where current levels of taxanes in the blood and their correlation with symptoms of acute intoxication are required.

## 2. *Taxus* spp. and Biologically Active Compounds

### 2.1. Overall Distribution of Compounds

Regarding the distribution of taxanes between individual plant genera, the content of paclitaxel (PTX), baccatine III (BAC III), 10-deacetyl paclitaxel (10-DAT) and 10-deacetylbaccatine III (10-DAB III) was compared in *T. media*, *T. mairei* and *T. cuspidata*. It was found that paclitaxel is most represented in *T. cuspidata* and least in *T. mairei*. Baccatine III is the most abundant in *T. cuspidata* and the least in *T. media*. 10-Deacetyl paclitaxel is most represented in *T. cuspidata* and least in *T. mairei*, while 10-deacetylbaccatine III most in *T. mairei* and least in *T. media* [23]. Poupat et al. (1999) compared the content of paclitaxel and 10-DAB III in the leaves of individual trees. They found that the lowest content of paclitaxel and 10-DAB III in *T. breviolia* is 0–15 and 77.4–297.6 µg/g, respectively. The highest content of paclitaxel was 0–169 µg/g in *T. cuspidata*, and the highest content of 10-DAB III in *T. wallichiana* was 247.6–594.9 µg/g. The content of paclitaxel decreased in the order *T. cuspidate* > *T. canadensis* > *T. floridana* > *T. wallichiana* > *T. brevifolia* [24]. Based on Table 1, it is possible to see that paclitaxel is most represented specifically in the tree part of *T. cuspidata* (46–1670 µg/g), *T. media* (15–1220 µg/g), *T. mairei* (<100–660 µg/g), *T. chinesis* (15–90 µg/g), *T. yunnanensis* (84 µg/g) and the least paclitaxel contains *T. fuana* (27 µg/g). For BAC III, the trend is as follows: *T. cuspidata* > *T. mairei* > *T. media* > *T. yunnanensis* > *T. chinesis* > *T. fuana*. Regarding the sum of concentrations of substances PTX, BAC III, 10-DAT, 10-DABIII, 7-epi-10-deacetylpaclitaxel (7-epi-10-DAP), cephalomannine (CEPH), 7-epi-deacetylbaccatine (7-E-DAB), 7-xyl-10- deacetylpaclitaxel (7-xyl-10-DAT), 7-epitaxol in the tree part of the selected trees, the highest concentration of these substances are in *T. cuspidata* (5137.4 µg/g), *T. media* (3502.2 µg/g), *T. mairei* (2540 µg/g), *T. chinesis* (2036.2 µg/g), *T. yunnanensis* (1124 µg/g) and the smallest concentration in *T. fuana* (322 µg/g) (Table 1). Comparing the total maximum concentration of taxanes in the needles of *T. chinesis* and *T. baccata* trees, 1964 µg/g and 893 µg/g was found, respectively, which is a more than two-fold difference (Table 1). Concentration of paclitaxel in woody parts is at most 10 times higher than in needles and more than 300 times higher than in red arils. For DAB III, it was observed that the highest concentration was in woody parts, then in needles, and the lowest in red arils. This trend can also be observed for other taxanes. Individual *Taxus* trees could be distinguished on the basis of some characteristic substances present in the given species or based on the given concentrations. There is high inter- and intra-species variability in taxane content [25,26], and it has been shown that this variability can depend on the season and taxoid type. The concentration of 10-DAB in first- and second-year needles of *T. baccata* grown in a Polish garden was highest in January and lowest in April [3], while in a study in Tehran, Iran [6], the highest concentration in yew needles was reached in August. The bark of *T. baccata* grown in the forests of the Czech Republic showed that the highest concentration of PTX and three other taxanes was in October and the lowest in January [27]. 

### 2.2. Physiochemical Properties of Biologically Active Taxanes

Taxanes belong to the most diverse group of pharmaceutically important natural tricyclic diterpenes [54]. They are formed in gymnosperms (*Gymnospermophyta*) belonging to the order Cupressales, family Taxaceae and species *Taxus* with genera such as *T. baccata* (English or European yew), *T. brevifolia* (Pacific or Western yew), *T. canadensis* (Canadian yew), *T. celebica* (Chinese yew), *T. cuspidata* (Japanese yew), *T. floridana* (Florida yew), *T. globosa* (Mexican yew), *T. wallichiana* (Himalayan yew) and *T. fuana* (West Himalayan yew). Wang et al. comprehensively described the structures of taxanes and divided them into 11 types according to their carbon ring system (6/8/6, 6/5/5/6, 6/10/6, 5/7/6, 5/6/6, 6/12, 6/8/6/6, 6/5/5/5/6, 5/5/4/6/6/6, and 8/6). According to the order of their discovery, they are divided into 10 types of taxane skeletons (I-X), and based on skeleton/substitution patterns, they were divided into 28 groups representing more than 600 isolated substances from various yews [40,55]. From this huge number of substances, it is obvious that their structure is very variable, and their side chains will play important roles. Among these secondary metabolites with anticancer activity, the most pharmacologically important is taxol (Figure 1), also known as paclitaxel (PTX) (5β, 20-epoxy-1,2α, 4,7β, 13α-hexahydroxytax-11-en-9one-4,10-diacetate-2-benzoate-13 ester with (2R,3S)-N-benzoyl-3-phenylisoserine) [56]. The core of taxol’s structure is a skeleton motif (6/8/6), which has several functional groups including two OH groups, one benzoyl group, two acetyl groups and an oxetane ring (Figure 1). Taxol itself as well as its side chains have been intensively studied, while we will only mention the biological activity of individual structures in general. Mazumder et al. reported that the acetyl group at C-10 is responsible for the variability of taxol analogs. The substituent at C-13 (it is H in paclitaxel, and an OH group in Baccatin III, Figure 1), the OH group at C-1, and the substituent at C-2 are significant variables for bioavailability. The oxetane ring (D-ring) is essential for enhanced activity [57]. Hao et al. describe that affinity for P-gp (P-glycoprotein, MDR1 and ABCB1) is largely dependent on the number and strength of hydrogen bonds in a compound. If C-10 is modified to PTX or Taxol C (the same as taxol with a small change to the C-13 substituent), the interaction with P-gp is reduced, and thus, they are not effluxed through the blood–brain barriers or out of the tumour cell [58]. Wang et al. describe that the 6/8/6 membered ring and free hydroxy group at C-2 are important for PTX activity. C-13 is important for bioviability (see above), but at the same time, removal of the side chain abolished its antimitotic and antimicrotubule activity. Although C-7 and C-10 in PTX interact with tubulin, C-13, C-2 and C-4 are more important for this interaction [40]. Most recently, Jara et al. summarized that the taxane ring, C-13 side chain, D-ring and 2’ position of the hydroxyl group and the hemochiral ester chain are essential for its correct antitumoral activity [4]. Matesanz et al. [59] observed that hydroxyl groups at C-7 and C-10 in cephalomannine (Figure 1) interact with microtubules, but are not essential for the antitumor activity in PTX [4]. 

Paclitaxel is a white to off-white crystalline powder. It is insoluble in water (5.56 × 10^−3^ g/L), with a melting point near 216 °C, and it is highly lipophilic. In addition, Taxol is unstable in many commonly used solvents and at temperatures above 60 °C. One of the problems was also related to its low solubility in water (poor oral bioavailability) and the fact that the drug was isolated in a very low yield [60]. Low solubility in water and high lipophilicity are related to its polar surface area (PSA). Table 2 describes the PSA values of PTX and some taxanes (134–221 Å), indicating their different solubility in water. In addition, synthetic/semisynthetic taxanes have a significantly higher PSA (222.88 ± 23.14 Å), which is also related to their poor oral bioavailability [58,61]. That is also why the development of various methods of administration to improve oral bioavailability (drug delivery transport) began, for example by binding to various carriers, different formulations arise, such as Cremophor EL (solvent based formulation of PTX), nab-paclitaxel (nab-P, albumin-bound nanoparticle formulation of PTX), polymer micellar PTX (monomethoxy poly(ethylene glycol)– block–poly(D,L -lactide) polymer formulation of PTX) and micellar formulation of paclitaxel encapsulated in the proprietary retinoid compound XR-17 (mixture of two iso-forms of N-retinoyl-L-cysteic acid methyl ester sodium salt) [4,10,62]. 

### 2.3. Method of Analysis in Taxus spp.

Advances in research related to the development and improvement of therapeutic procedures have led to a growing interest in analytical methods aimed at identifying and quantifying molecules in various plant parts of the *Taxaceae familia*. These methods should be relevantly optimized to achieve the highest possible sensitivity and specificity for the analytical signal of interest, lower consumption of extraction reagents or the fastest possible analyses. Simplification of pre-analytical sample processing and faster analyses are beneficial for reducing error sources in method validation and at the same time are necessary for current monitoring of taxane levels in clinical studies or the search for new biologically active substances [63,64]. Using these recent simplifications, the gentlest methods are supercritical fluid extraction (SFE) and ultrasound-assisted extraction (UAE). Although SFE uses non-toxic solvents (water or carbon dioxide), its industrial applicability is hindered by the use of complex, expensive equipment and relatively low product yields. Alternatively, UAE combines ultrasound treatment and conventional Soxhlet extraction (SE) [65] with solid/liquid systems being the most common candidates for UAE. One of its main advantages lies in the enhancement of the mass transfer brought about by acoustic-induced cavitation in a liquid medium [35,66]. This mechanism allows researchers to use a reduced amount of organic solvents compared to Soxhlet extraction. 

In the literature, several analytical methods have been described for the quantification of taxanes in various matrices. These methods include, for example, immunoassays and micellar electrokinetic chromatography (MEKC). However, most techniques involve the use of high-performance liquid chromatography (HPLC) coupled with an ultraviolet (UV) or mass spectrometric (MS) detector due to higher sensitivity and resolution, as well as the ability to work with different types of matrices (Table 1). In addition, the HPLC technique is used as a standard method in the pharmaceutical industry. Besides HPLC, gas chromatography (GC) and gas chromatography–mass spectroscopy (GC-MS), Fourier-transform infrared spectroscopy (FTIR) and/or nuclear magnetic resonance (NMR) are also used (Table 1). Their disadvantage is that the development of new methods focuses mainly on the detection of biologically active substances in extracts from the *Taxaceae familia*, while analytical procedures for the detection of these substances in human biological samples are rarely developed (See Section 4). 

The fastest and the most sensitive method for taxane analysis published in the last 5 years (Table 1) is the ultra-performance liquid chromatography with tandem mass spectrometry (UHPLC-MS/MS) method, which was developed for the simultaneous determination of seven taxanes and seven flavonoids in twigs and leaves of three species from the *Taxaceae familia*. The LOQs of all analytes ranged from 0.01 to 1.66 ng/mL. The method had the best validation parameters of all the published works described in present paper, with the extraction taking just over 20 min and the analysis 5 min [67]. 

As can be seen from Table 1, the most monitored plants in the last 5 years were *T. cuspidata*, *T. chinesis* and *T. baccata*. Six articles also described the analytes found in *T. media*. As for the plant parts, leaves are the most researched, with up to 13 articles devoted to them. In addition to leaves, stems, twigs, red arils, sapwood, heartwood, essential oil, roots and branches were analysed. Carotenoids, amino acids, taxanes, vitamin C, polyphenols, macroelements, microelements, antioxidants, flavonoids, fatty acids, proteins, lipids, carbohydrates, xanthophyll esters were observed in red arils. Flavonoids, phenols, polysaccharides and, last but not least, mainly taxanes were observed in the needles. Various ethers, alcohols, acids, flavonoids, ketones, phenols, esters were monitored in the tree part. Aldehydes, pyridines, fatty acids, volatile substances, alkanes, lipids, nucleotides, amino acids, alkaloids, vitamins, terpenes, cofactors, carbohydrates, phenylpropanoids, hormones, antioxidants and taxanes were described in twigs. The most frequently used analytical methods were liquid chromatography, either high-performance liquid chromatography with ultraviolet detection (HPLC-UV), high-performance liquid chromatography with diode-array detection (HPLC-DAD), liquid chromatography with tandem mass spectrometry (LC-MS/MS), liquid chromatography with mass spectrometry (LC-MS), ultra-performance liquid chromatography with tandem mass spectrometry (UPLC-MS/MS), liquid chromatography with quadrupole time-of-flight mass spectrometry (LC-QTOF), or high-performance liquid chromatography with photodiode array detection (HPLC-PDA). This was followed by the use of the gas chromatography–mass spectrometry (GC-MS) method; the least used were ultraviolet–visible spectroscopy (UV-VIS), Fourier-transform infrared spectroscopy (FTIR), gas chromatography with flame ionisation detector (GC-FID), thin-layer chromatography (TLC), high-resolution mass spectrometry (HR-MS) and nuclear magnetic resonance (NMR). The distribution of individual taxanes in different parts of the plant is not uniform. The concentration of taxanes in red arils was in the range 3.9–38 µg/g for 10-DAB III; BAC III 1–28 µg/g; CEPH 0.04–7.2 µg/g; taxinine M 0.02–2 µg/g; paclitaxel (PTX) 0.02–5.5 µg/g. In needles, the concentration range was as follows: 10-DAB III 21.1–703.4 µg/g; PTX 9.31–162.75 µg/g; 10-diacetyltaxol 14.7–236.49 µg/g; BAC III trace-120.48 µg/g; CEPH 19.2–105.69 µg/g; 7-xyl-10-DAT 351.44–546.95 µg/g; 7-epi-10-DAT 94.45–166.44 µg/g. The highest concentration range for paclitaxel 30–1670 µg/g was observed in woody parts: BAC III 5–800 µg/g; 10-DAP 100–850 µg/g; 10-DAB III 50–1000 µg/g; 7-E-10-DAP 170–300 µg/g; CEPH 30–150 µg/g. 

Monitoring the composition of Taxaceae plant species can be helpful in the search and isolation of new biologically active substances, precursors for the synthesis of new drugs and, last but not least, for the diagnosis of intoxication. 

## 3. Pharmacology of Taxanes

### 3.1. Molecular Mechanism of Action

The cytoskeleton is a strictly and specifically arranged system of filaments, which is involved in cell division, intracellular movement of organelles and building cellular materials, and last but not least, maintains the necessary shape of the cell. We know three types of filaments: intermediate filaments, microtubules and actin filaments. Microtubules ensure the position of membrane-enclosed organelles, control intracellular transport and play an essential role in cell division. They have the shape of long hollow cylinders formed by the protein subunit tubulin, which is a heterodimer composed of α-tubulin and β-tubulin (α-β interaction). They connect vertically to each other so that the top of the β monomer in the heterodimer is connected to the bottom of the α monomer in the next heterodimer to form a single-chain protofilament (β-α interaction). Protofilaments arise in different numbers in different cells. Mostly thirteen protofilaments form through lateral α-α and β-β interactions in the cylindrical hollow structures—microtubules [68,69,70]. Monomer α-tubulin permanently binds guanosine-5’-triphosphate (GTP) at the dimer interface. Monomer β-tubulin binds GTP and guanosine-5’-diphosphate (GDP), which are important in the growing (polymerisation) and shrinking (depolymerisation) of microtubules, respectively. If the concentration of heterodimers with bound GTP or GDP is higher or lower than the critical concentration, microtubule polymerization or depolymerisation occurs, respectively. The critical concentration of tubulin may vary between species. For example, mung tubulin is up to 10 times lower than for brain tubulin [70]. The end of microtubules, where the polymerisation and depolymerisation of microtubules is more dynamic, is called plus (+), the other end is then minus (-). Natural taxanes such as paclitaxel (PTX) or cephalomannine (CPH) bind in the cavity of this cylindrical structure (called the nucleus or lumenal site), which prevents the polymerisation of microtubules (in “freezing” effect, stabilising them “kinetically”). [59]. These taxanes are also used as microtubule-stabilising agents in the study of the pathway of microtubule assembly. Matesanz et al. described in detail the luminal site and pore site as model binding sites for taxanes. These are bound in these places via substituents in the C-7 and C-10 positions, increasing or decreasing the interprotofilament angle, respectively [59]. Binding affinity in these paclitaxel binding sites plays a key role for the biological effect of taxanes and is at the same time a useful tool for detecting the pharmacological effect of potential chemotherapeutics-taxane analogs/derivatives. In vitro measured apparent binding constant (K_b_) for PTX is 1.43 ± 0.17 × 10^7^ M^−1^ [71] and 3.7 × 10^7^ M^−1^ [72]. In vivo experiments measured the binding equilibrium dissociation constant (K_d_) to be significantly higher in RPE1 cells K_d_ = 5 × 10^9^ ± 0.1 M [73], in HT1080 cells K_d_ = 7.6 ± 0.9 × 10^9^ M [74] and 5.0 × 10^9^ in MCF7 cells [75]. CPH is K_b_ = 0.69 ± 0.08 × 10^7^ M^−1^ [73]_,_ and Baccatine III is K_b_ = 1.5 × 10^5^ M^−1^ [72] for the PTX binding site. Pineda et al. in the cell lines RPE1 and HT1080 found that loss of microtubule dynamics, micronucleation and mitotic arrest were approximately at 0.1 (10%), 0.2 (20%) and 0.6–1 (60–100%) occupancy of the binding site for PTX, respectively. At the same time, these authors state that in order to completely stop tumour growth in a mouse xenograft tumour model, it is necessary that at least 80% of the binding sites for PTX on microtubules were occupied [74]. However, the binding affinities constants are determined at different temperatures by different methods and in different ways (in vitro or in vivo). The pharmacological effect of a ligand on its binding site can be influenced to some extent by the presence of a drug efflux pump [76]. Kuh et al., however, found that when Pgp-mediated paclitaxel efflux in MCF7 cells is inhibited, PTX passes through by free diffusion, and compared to extracellular binding, the dissociation constant for intracellular binding is approximately 160 times smaller, while the maximum binding capacity is 15 times larger [75]. 

### 3.2. Human Pharmacokinetic Parameters of Taxanes

The effect of the drug on the organism should be therapeutically monitored using basic pharmacokinetic parameters (PK), for example its maximum concentration in the blood (C_max_) [77]. However, an even more important parameter is its concentration in the local therapeutic site, in the case of taxanes inside the tumour cell. Only a few studies are describing intracellular concentrations of paclitaxel (PTX). Kuh et al. found that when MCF7 was incubated with PTX in the concentration range of 1–5000 nM, there was a high intracellular-to-extracellular concentration ratio, while after 4 h, the intracellular concentration was the highest (0.31–269,010 ng/mL). Only treatment with 85.4 and 854 ng/mL PTX enhanced the polymerisation significantly, and after 24 h, induction of tubulin production was also observed [75]. In another study, Wang et al. investigated the effect on cycle arrest, tubulin assembly and cell viability when HeLa cells were incubated with 0–85.4 ng/mL AbraxaneTM (albumin nanoparticle-based), further with PGA-PTX, also known as CT-2103 (polymer nanoparticle-based degradable paclitaxel carrier, with poly(L-glutamic acid)-paclitaxel conjugate) and pure PTX. For Abraxane and pure PTX, most cells were arrested in the G2 phase (concentration about 8540 ng/mL) what correlated with inhibiting 50% tubulin assembly. The percentage of viable cells decreased with the increasing number of polymerised tubulin, while the IC_50_ for Abraxane was 6.9 ng/mL, and for PTX, the IC_50_ was 11.44 ng/mL. PGA-PTX needed a longer incubation time (72 h) to obtain similar kinetics as the previous two PTX formulations, which was related to its much lower concentration of free PTX in the cell lysate (20 ng/mg protein vs. approximately 80 ng/mg protein). The IC_50_ for PGA-PTX was 9.4 ng/mL at prolonged exposure to HeLa cells [78]. Determination of the mechanisms of action of PTX at the cellular level, or determination of the intracellular concentration at a known extracellular concentration, represents a direct insight into its effect and can help in clinical studies application. 

It is obvious that the pharmacokinetic parameters for PTX will be different (Table 3) if administered in different formulations, alone (monotherapy) or together with other drugs. By reviewing 53 pharmacokinetic studies, Stage and co-authors found non-linear pharmacokinetics where Cremophor EL^®^ and nab-paclitaxel were administered weekly and in 3-week dosing cycles. The time of PTX level in plasma above the threshold effective value (0.05 µmol/L = 42.7 mg/mL) was 23.8 h. For Cremophor EL^®^, clearance (CL) decreased with increasing dose, in the shorter (1, 3 and 6 h) and in the longer infusion times (24 and 96 h). C_max_ values increased with increasing dose at both infusion times and reached higher values at short infusion. A similar dose-dependent trend in CL and C_max_ was observed with the nab-PTX formulation [79]. 

Another study compared the PK parameters of nab-PTX with micellar formulation-PTX, which is formulated with the surfactant XR17 (mixture of two iso-forms of N-retinoyl-L-cysteic acid methyl ester sodium salt) (Table 3). For 28 patients with metastatic breast cancer (Caucasian females, average BSA was 1.7 ± 0.2 kg/m^2^), the same dose of 260 mg/m^2^ was administered (1 h intravenous infusion) for both formulations. Micellar PTX and nab-PTX were considered clinically equivalent. Unbound plasma PK profile of PTX reflected above-mentioned total PK profile with concentrations of approximately 5% of the corresponding total concentrations [80]. Clearance was not investigated, but similar PK parameters were observed earlier in a different group of patients (Caucasians) with solid metastatic tumour. CL was approximately 15 L/h/m^2^, unchanged over the dose range (150–275 mg/m^2^), and C_max_ along with area under the plasma concentration–time curve extrapolated to infinity (AUC_inf_) increased linearly over the dose range [81]. PTX protein-bound particles for injectable suspension and Abraxane^R^ were considered as clinically equivalent. 

Chinese patients (28) with breast cancer (average body surface area (BSA) = 1.63 kg/m^2^) received a dose of 260 mg/m^2^ PTX of two formulations in the form of an infusion lasting 30 min. For both formulations, T_1/2_ (24.48 ± 4.47 vs. 27.32 ± 10.10 h) was approximately the same [82]. Chinese patients had significantly higher C_max_ compared with the studies described thus far (Table 3). This could be due to the shorter duration of the infusion (30 min) with approximately the same formulation and amount of 260 mg/m^2^/1.63 kg/m^2^ = 160 mg/kg compared to the study by Borg et al. [81] with a dose of 260 mg/m^2^/1.7 kg/m^2^ = 153 mg/kg (duration of infusion 1 h). This was observed earlier when summarizing 53 PK studies [79]. Furthermore, differences in C_max_ may be related to ethnic characteristics because at a dose of 260 mg/m^2^ during a 30 min infusion, European/Americans had significantly higher C_max_ (19,556 ng/mL, not shown in the table) [83]. Tumour concentrations of PTX are much lower than plasma concentrations, and thus, it can be assumed that tumour type and size would not affect plasma concentration in the above studies. Pharmacokinetics of PTX in tumour cells is poorly described in the literature. Desai et al. found that in the xenograft tumour, the highest concentration of PTX formulations was around 1 h after a dose of 20 mg/kg and reached a value of approximately 7 ng/mg [84]. At a dose of 50 or 100 mg/m^2^ PTX together with carboplatin or concomitant radiotherapy to fourteen patients (median body surface area, BSA = 2.1) once per week with an infusion duration of 1 h, it was found that after 4 h from the dose, the concentration of PTX inside the tumour (oesophageal tumour) was 2.91 ng/mg. This was significantly higher than in its surroundings (oesophageal mucosa = 2 ng/mg). Total amount of active drug that reaches the systemic circulation (AUC 0–48 h, 1500–10,000 ng·h/mL) and the concentration taken 4 h (50–350 ng/mL) after the dose were positively correlated with the intratumoral concentration (approximately 1.5–7.0 ng/mg). The concentration in the oesophageal mucosa did not correlate with these parameters and was approximately the same at the beginning and at the end of the PTX dosing cycle [85]. 

The following studies were unique in their parameters, for example by examining the relationship between steady state concentration (C_ss_) and antitumor response. C_ss_, AUC, and total body clearance CL were analysed in 48 patients at a dose of Cremophore EL 250 mg/m^2^ by 24 h infusion every 21 days (Table 3). The high frequency of hematologic effects prevented the investigation of the relationship between C_ss_ and antitumor response, and as the authors concluded, the solution would be lower doses [86]. 

Similarly, Rowinsky et al. in 1999 stated that with C_ss_, the binding sites may already be saturated and the trend between concentration and effect on the organism could only be observed at low doses. The dose was 75 mg/m^2^ cisplatin i.v., combined with either a low dose of paclitaxel (135 mg/m^2^, 24-h i.v., infusion) for 78 patients or a higher dose of paclitaxel (250 mg/m^2^ i.v., 24-h i.v.) for 87 patients with granulocyte colony-stimulating factor. The authors observed the lack of relationships between disease outcome, and both paclitaxel dose and C_ss_ were obtained. Moreover, no relationship was observed between C_ss_ and the tendency to develop neurotoxicity, but the relationship with the development of the worst degree of leucopoenia was significant [87]. 

Hurria et al. found that C_max_ increases with patient age in 40 patients (mean age 60, (30–81) years) with metastatic breast cancer. However, there was no significant difference in toxicity, dose reduction, or dose omissions in older versus younger adults [88]. 

**Table 3 ijms-23-15619-t003:** Human pharmacokinetic parameters at different PTX formulations.

PTX Formulation	Dose (mg/m^2^)	C_max_ (ng/mL)	T_max_ (h)	CL	AUC (h ng/mL)	Refs.
Cremophor EL^®^ nab-paclitaxel	175 (135–240)	median 4335 (2635–8160)	na	median 12 L/h/m^2^	na	[79]
Micellar PTX	260	8006 ± 1703	50 ± 17	na	13,484 ± 3491 (0–10 h)	[80,81]
nab-paclitaxel		8073 ± 4124	46 ± 20	na	11,388 ± 3123 (0–10 h)	
PTX Abraxane^®^	260	12,295 ± 2370 12,771 ± 2065	0.5 0.5	na	12,587 ± 2932 (0-inf) 13,078 ± 2880 (0-inf)	[82]
Cremophore EL^®^	250	725.9 ± 179.34 (steady state)	na	256 ± 72 mL·min^−1^·m^−2^	17,336.2 ± 4355.4	[86]
cisplatin + PTX cisplatin + PTX	75 + 135 75 + 250	273.3 (102–2306); 691 (93.4–3074.4),	na	na	na	[87]
nab-PTX	100	5482 ± 3967	na	37.16	5316 (3001) (0–48 h)	[88]

### 3.3. Human Pharmacodynamic Parameters of Taxanes

The most well-known substance from the group of taxanes used as a medicine is taxol A (PTX). It is an antineoplastic substance that is used for first-line therapy and for the subsequent therapy of advanced tumours [4]. The mechanism of action of this drug is non-selective for human cells, but rapidly dividing cells (tumour, hematopoietic, intestinal, skin, etc.) are most affected by this treatment, which can be observed in the change of progression in the tumour disease as well as in the unwanted effect on organ systems. Its mechanism of action is on microtubules [4], and for example in canine mammary carcinoma, it was observed that increasing concentrations of PTX decreased cell viability, increased the percentage of cell apoptosis and increased relative reactive oxygen species (ROS) levels [89]. This causes cell cycle arrest and subsequent cell death. Another antitumor mechanism of paclitaxel is the induction of calcium depletion in the cell mitochondria, which leads to the release of apoptogenic factors and the initiation of apoptosis [90]. 

For monitoring the drug effect on the body, the dosage strategy (dose, length of administration of the infusion dose, total number of doses, time intervals between individual doses) is important, followed by its maximum concentration in the blood, which should reflect the observable pharmacodynamic effects on the body. Administration of PTX in weekly doses prolongs progression-free survival compared to dosing in three-week intervals but does not improve overall survival, while the incidence of adverse effects is slightly higher [91]. For example, weekly dosing (6 cycles) in ovarian cancer therapy uses a dose of 80 mg/m^2^ PTX in combination with carboxyplatinum, a three-week dosing schedule uses 175 mg/m^2^ (6 cycles) [92], and higher incidence of adverse effects in terms of anaemia and neuropathy was observed. A similar trend can be observed in the treatment of patients with metastatic breast cancer [93]. The results of clinical studies focused on the influence of the dosing strategy on the occurrence of adverse effects are very heterogeneous. Research activity in this area is very extensive. Dosing of taxanes in individual cytostatic schemes during the treatment of various cancer diseases in different disease stages is a complex issue that exceeds the possibilities and scope of this publication. 

#### 3.3.1. Common Physiological and Unwanted Effects

Most patients treated with paclitaxel experience adverse effects, the incidence and severity of which depend on the specific composition of the drug, including excipients, patient risk factors, and the chosen cytostatic therapy strategy [94]. Due to the mechanism of action of paclitaxel or taxanes on microtubules, which results in the inhibition of the cell division cycle, in addition to tumour cells, organ systems containing rapidly dividing cells are most often affected, such as: the hematopoietic system and the development of myelosuppression, the neurological system and the development of neuropathy, the gastrointestinal system and the development of the upper and lower dyspepsia, diarrhoea and skin leading to alopecia. The incidence of adverse effects and symptoms of toxicity in the treatment of cancer patients with paclitaxel ranges from 70% to 83.6% [95,96]. Due to the frequent occurrence of adverse effects, which can limit the dosing schedule of paclitaxel, the effort is to find suitable combinations with other substances in order to reduce them. Curcumin appears to be one of the investigated natural substances that can reduce the incidence of unwanted effects [97]. 

#### 3.3.2. Hypersensitivity Response and Anaphylaxis

Although the administration of paclitaxel is generally safe and well tolerated, cases of hypersensitivity reactions up to fatal anaphylaxis may occur. The occurrence of a hypersensitivity reaction during the administration of paclitaxel is approximately 10% of patients, despite the administered prophylactic therapy with antihistamines and glucocorticoids [98]. The mechanism of the hypersensitivity reaction is not fully known and may occur immediately after the first administration, without prior sensitisation [99]. The hypersensitivity reaction is mainly caused by adjuvants that allow for the transport and transfer of hydrophobic paclitaxel into cells. The most frequently used carrier solution for the drug is polyoxyl 35 castor oil (Cremophor EL^®^), which acts as an activator of basophils and mast cells and as a direct histamine liberator. Furthermore, allergic or even anaphylactic reactions occur through IgE- and IgG-mediated immune response to paclitaxel and/or adjuvants [98,100]. For this reason, there is a high need for the development of another carrier vehicle for the drug, which will enable a more efficient transfer into the tumorous tissue and reduce the occurrence of adverse effects. The trend in development is the creation of suspensions based on nanoparticles, while nanoparticles can be based on human albumin (nab-paclitaxel, FDA-approved), liposomes (EndoTag-I, LEP-ETU), micelles (Genexol-PM), nanodispersions (substance PICN) and microparticles (substance AI-850) [101]. Nab-paclitaxel has a 33% higher penetration into tumorous tissue compared to conventional paclitaxel [101]. According to a meta-analysis by Chou et. al, who analysed the occurrence and degree of severity of hypersensitivity reaction when using different paclitaxel solutions based on nanoparticles, the liposomal phospholipid emulsion has the lowest incidence of hypersensitivity reactions, in which a lower dose can be administered to achieve the same therapeutic effect [94,101]. A colloidal suspension of albumin-paclitaxel nanoparticles (nab-paclitaxel) created for the purpose of increasing paclitaxel permeation through tumour albumin receptors proved to be 4.5 times more effective compared to conventional PTX [93,102]. An immediate hypersensitivity response usually occurs during the first cycle of drug administration, minutes after the start of the infusion. This response is manifested by flushing, itching, rash with back and chest pain [103]. If hypersensitivity reaction progresses to a moderate severity, symptoms of dyspnoea are added to the symptoms without a decrease in blood oxygen saturation and without a decrease in blood pressure [104]. In a severe stage of hypersensitivity reaction, the reaction is accompanied by hemodynamic instability, a decrease in blood oxygen saturation, nausea and vomiting. Considering that hypersensitivity response or anaphylactic reaction are potentially life-threatening conditions, it is recommended to start premedication with H1, H2 blockers and dexamethasone before administration of standard paclitaxel [98]. Despite the administered premedication, cases of fatal hypersensitivity reactions have been described [99]. Due to the fact that the administration of standard paclitaxel is associated with a hypersensitivity reaction, it is necessary to monitor vital functions during administration and to immediately stop administration at the appearance of the first symptoms. Nab-paclitaxel is to date the only FDA-approved preparation containing taxane-saturated polymer nanoparticles of human albumin non-covalently bound to paclitaxel, which does not require premedication with H1, H2 and glucocorticoids [105]. 

#### 3.3.3. Neurotoxicity

Neurotoxicity is one of the first and most common symptoms of chemotherapy-related toxicity. The most common first manifestation of neurotoxicity is the development of paresthesis; then, as the symptoms of toxicity progress, sensory disorders, motor disorders, and autonomic nervous dysregulation occur. Primarily, the first symptoms occur on the distal parts of the limbs and progress proximally [96]. The emergence of this peripheral neuropathy is explained by the arrest of the growth of distal part of the axons of sensory neurons in the skin and proximal parts of the limbs, which causes the sensory symptoms of neurotoxicity described above [106]. Taxanes accumulate in the ganglia of the posterior spinal horns, which causes initial disturbances in axonal transport and changes in Schwann cells, which subsequently lead to ganglion neuropathy and axonopathy. Symptoms of neurotoxicity are a frequent reason for the need to reduce the administered dose of paclitaxel or even to interrupt its administration. The rate of neurotoxicity depends on the type of paclitaxel administered. In the study realised by Guo et al. in 2022, it was found that paclitaxel administered in the form of albumin-paclitaxel nanoparticles, nab-paclitaxel, has a higher anti-tumour effect and a lower incidence of drug resistance, thereby increasing the survival of patients with breast cancer, but on the other hand, it also has a higher incidence of neurotoxic symptoms compared to standard paclitaxel. Atypical symptoms of neurotoxicity include numbness, burning sensations, firing of discharges, hyperalgesia, acute pain syndrome, convulsions, and transient encephalopathy. Neurotoxicity and its manifestations significantly reduce the quality of patient’s life and can lead to paralysis, withdrawal from normal life, and disability. The incidence of nab-paclitaxel toxicity is 70%, and 10% of patients experience a severe grade with significantly reduced quality of life. Symptoms begin to appear during the first cycle of administration and may persist for up to one year [96]. The incidence of taxane-induced neurotoxicity occurs in 13–62% of cases, with severe grade occurring in 5–10%. In cases where patients were treated primarily with taxanes for primary tumour, they showed signs of peripheral neurotoxicity for a long time, and these symptoms significantly reduced their quality of life [107]. Cryotherapy with frozen gloves and socks can be used as a treatment method that has promising results in preventing the occurrence of neuropathy on acres [108]. 

#### 3.3.4. Gastrointestinal (GIT) Toxicity 

Among the most common symptoms of this type of toxicity are nausea, vomitus and diarrhoea, but adverse effects can also lead to the development of severe damage to the GIT, the development of colitis and intestinal perforation. Paclitaxel has a low emetogenic effect and is associated with approximately a 10–30% risk of nausea. The mechanism of toxicity is attributed to the direct binding of the ligand (PTX) to the mucosal Toll-like receptor 4, which mediates damage to the intestinal epithelium [109]. Toll-like receptor 4 is a transmembrane receptor that is found on cell surfaces of macrophages, dendritic cells, and neutrophils, as well as on the surface of epithelial cells lining the respiratory and intestinal tracts. It triggers the inflammatory cascade by activating pro-inflammatory genes [110]. Risk factors for nausea include combination with other more emetogenic chemotherapeutics, younger age, female gender, low alcohol consumption, previous history of motion sickness and gestational nausea. Diarrhoea occurs in approximately 50% of patients, and the mechanism of this toxicity consists in the disruption of the villous part of the enterocyte, disruption of the integrity of enterocytes and an increase in their permeability to toxins [111]. The incidence of taxane-induced colitis is not frequent. In a study by Chen et al. in 2019, the occurrence of taxane-induced colitis was documented in 0.2% of cases, while the development of this disease was a gradual deterioration of GIT functions over time. Treatment with nab-paclitaxel resulted in the development of a more severe degree of colitis compared to conventional paclitaxel, with 100% of nab-paclitaxel-treated patients developing colitis requiring hospitalisation, compared to 34% in case of standard paclitaxel-treated patients [112]. 

#### 3.3.5. Cardiovascular Toxicity

The cardiotoxicity of taxanes has been known for a long time and occurs at a higher incidence when paclitaxel is administered with other cardiotoxic chemotherapeutics. The exact mechanism is unclear, but several pathophysiological mechanisms have already been identified [113]. A hypersensitivity reaction causes the release of histamine into the bloodstream, which subsequently causes stimulation of the histamine receptors of the myocardium and a change in the conduction system of the heart, which leads to arrhythmias [114]. Bradycardia, as the most common symptom of cardiotoxicity, arises on the basis of central autonomic dysregulation, which is a manifestation of neurotoxicity [115]. The proarrhythmogenic effect is also caused by the interaction of PTX with HERG K channels (human ether-a-go-go), which causes a rapid decrease in the influx of potassium into the cardiomyocytes and causes a prolongation of the QT interval [116]. Another probable mechanism of action of taxanes lies in a direct effect on cardiomyocytes by causing an increased formation of reactive oxygen species in mitochondria, which leads to their damage, and the permeability of mitochondrial membranes increases, which causes cardiomyocyte dysfunction and subsequent myocardial dysfunction [113]. The incidence of taxane cardiotoxicity according to the analysis of registers of adverse effects carried out by Batra et al. is approximately 11.8% [117]. The most common symptom was the occurrence of arterial hypertension requiring hospitalisation in 30.2% of cases. Taxane-induced cardiac symptoms were detected in 16.6% of cases [117]. Myocardial toxicity is manifested by acute or subacute reversible bradycardia, which is typical for paclitaxel, AV blockades, prolongation of the QT interval, atrial or ventricular fibrillation and left ventricular dysfunction [117,118]. Congestive heart failure is less common compared to other antimicrotubular agents such as vinca alkaloids, while in monotherapy, the incidence of cardiotoxicity symptoms is lower compared to combinations. The combination of paclitaxel with anthracyclines causes reduced elimination of anthracyclines in plasma, which increases their toxicity on the cardiovascular system [119,120]. Risk factors for the occurrence of these adverse effects are age, the presence of diabetes mellitus, hypertension and previous radiation therapy of the chest. It is recommended to avoid other combinations of drugs with a cardiogenic effect, such as tricyclic antidepressants or beta-blockers [90]. There is a lack of evidence on the myocardial effects of nab-paclitaxel, the incidence and severity of cardiovascular adverse effects compared to conventional paclitaxel. 

#### 3.3.6. Haematological Toxicity, Myelosuppression

One of the most common adverse effects of paclitaxel treatment is neutropenia and immunocompromitation. The mechanism of occurrence of this undesirable effect consists in the direct effect of the taxane on cessation of the division cycle of hematopoietic stem cells. Neutropenia caused by this mechanism is reversible, but directly dependent on the method of administration. Neutropenia is more common with 1 h paclitaxel regimens compared to 3 h regimens. Neutropenia and leukopenia are the most common adverse effects of therapy [90]. At the same time, anaemia and thrombocytopenia are also present in a mild form as a result of complex myelosuppression [121]. In a meta-analysis by Chou et al. in 2020, nab-paclitaxel was found to cause a higher incidence of neutropenia and leukopenia compared to conventional paclitaxel [94]. The treatment of this unwanted effect consists in the administration of colony-stimulating (CSF) or granulocyte colony-stimulating (G-CSF) growth factors [122]. Myelosupression is one of the serious adverse effects that threaten the patient with the development of a life-threatening infection [121]. 

#### 3.3.7. Respiratory System

The most common adverse effects of paclitaxel affecting the respiratory system are interstitial pneumonitis and capillary leak syndrome, which can develop within hours to weeks after therapy initiation. The most common occurrence of this complication is in patients with non-small-cell lung cancer, breast and prostate tumours [123]. Pneumonitis as an adverse effect of chemotherapy was defined as the presence of an otherwise unexplained worsening of dyspnoea in the last 30 days, the presence of ground-glass-opacities (GGO) or lung consolidations on high-resolution computerized tomography (HRCT), and excluded infectious origin, excluded progression of the tumour process to the lungs (in the lungs), absence of heart failure and no more than 4 weeks since the last dose of chemotherapy [124]. The mechanism of the development of this disease consists in a type IV immune response, a cell-mediated response, when an antigen is presented to T cells and causes their activation [125,126], while nab-paclitaxel also causes pneumonitis but at a lower incidence compared to conventional paclitaxel [127]. Another mechanism of pneumonitis development is the paclitaxel-activated release of tumour necrosis factor α (TNF-α) from macrophages, abnormal expression of cyclooxygenase 2 (COX-2) and initiation of a pro-inflammatory cascade that directly destroys the alveolar epithelium [128,129,130]. The incidence of this complication is approximately 1-5%, while the severity of the disease is severe to very severe; in the case of using nab-paclitaxel, it is moderate [127]. Patients who require the administration of G-CSF or CSF growth factors during paclitaxel therapy also have a higher incidence of interstitial pneumonitis [131]. Symptoms of pneumonitis include cough, dyspnoea, weakness, malaise, chest tightness, weight loss, and body aches [123,132]. Pulmonary fibrosis is described as a late complication of pneumonitis, which is irreversible damage to the lung parenchyma [133]. Risk factors are mainly pre-existing lung disease, chest radiotherapy, combination with other cytostatics, and taxane dosing strategy (doses higher than 250 mg/m^2^ are associated with a higher risk; weekly administration is riskier compared to triweekly administration) [133]. High-dose glucocorticoids [134], COX-2 inhibitors [128] and symptomatic therapy are used to treat this toxicity. 

#### 3.3.8. Skin Toxicity

Skin toxicity occurs in 89% of patients [135]. Paclitaxel causes many adverse effects on the skin through several mechanisms. It is directly cytotoxic to keratinocytes, causing allergic and hypersensitive reactions of type I and IV, which affect, among other parts, the skin [136]. Another mechanism of toxicity is inhibition of Bcl-2 expression, increase in p53 and p21 expression, decrease in vascular endothelial growth factor (VEGF), increase in markers of oxidative stress and proinflammatory markers, decrease in markers of elasticity and increased pigment production in melanocytes [137]. The most common taxane-induced skin symptoms typical of a hypersensitivity reaction are rash and pruritus, which are not serious. However, some patients may experience symptoms of hypersensitivity associated with systemic symptoms such as bronchospasm and blood pressure drop [104]. Skin toxicity is characterized by macular and papular skin changes that cause pruritus and pain. Painful, bilaterally occurring inflammatory patches in the axillary and inguinal regions can even have the character of toxic erythema [138]. The occurrence of eczema and folliculitis is also described [139]. Alopecia after administration of paclitaxel is present in at least 60% of patients and causes considerable distress to patients during chemotherapy, while 10% may develop permanent alopecia [140,141]. Toxic manifestations on the nails occur in up to 89%, most often in the form of onycholysis [140]. 

Cryotherapy can be used to prevent toxic manifestations on the skin and nails, having a good preventive effect and minimal side effects [108]. Lugtenberg et al. carried out research on the effect of cryotherapy on the prevention of alopecia during the administration of paclitaxel in monotherapy, while they found that cooling the head, which begins before the administration of paclitaxel and continues 20–45 min after administration, was effective and led to the prevention of severe alopecia in 78% of cases [142]. 

#### 3.3.9. Sex Differences in Toxicity

There are differences in the pharmacokinetics, occurrence of adverse effects and responses to administered taxanes between men and women. Men have a faster elimination of the drug compared to women, which predicts a lower incidence of adverse effects in men and at the same time alters the maximum tolerable doses within the treatment plan. Compared to women, men have a 20% higher elimination capacity. Paclitaxel-induced toxic manifestations are more frequent in women, the degree of toxicity is more severe and the duration of adverse effects is longer [143]. 

#### 3.3.10. Renal Toxicity

Administration of paclitaxel is nephrotoxic already at therapeutic doses. Tubular necrosis and, at higher doses, glomerular atrophy were observed. The mechanism of nephrotoxicity results directly from the effects of paclitaxel on the inhibition of the cell cycle and the occurrence of apoptosis, especially of proximal tubular cells [144]. 

## 4. Toxicity of Taxanes

### 4.1. Clinical (Non-Lethal) Toxicity

Intoxications with plants from the *Taxus* spp. are not so frequent but may also occur due to the easy availability of this plant. Intoxication with paclitaxel (PTX) is rarer, since it is an infusion dose applied in a hospital, and its availability is thus considerably limited. In addition to the usual unwanted effects associated with the treatment, several cases were observed where PTX or docetaxel (semi-synthetic taxane) was a cause of ethanol intoxication. In one case, PTX was administered by infusion of 600 mg (50 mL of absolute ethanol, equivalent to 1 L of beer) with symptoms of alcohol intoxication with a concentration of ethanol in the blood of 0.98 g/kg [145]. Other authors described that a dose of 300 mg (25 mL of ethanol, 0.5 L of beer) in three hours resulted in a feeling of head pressure, a change in balance, and ataxic gait with a wide base developed, which lasted 4–6 h [146]. During the first infusion of docetaxel (the dose was not given) to one patient with breast cancer, the treatment was interrupted when she was committed to the psychiatry ward because of a relapse of her alcohol addiction [147]. Another study describes a patient who had a ninth cycle of docetaxel 75 mg/m^2^ and who developed symptoms of alcohol intoxication such as blurred vision, drowsiness, imbalance, mild confusion and diplopia. According to the authors, the dose was 150 mg of docetaxel (ethanol inside amounts to 75 mL of beer) [148]. Although by default, the effects of alcohol are described with an oral dose, and thus, ethanol has a higher effect on the body in an infusion dose. The symptoms of ethanol described above are significantly higher due to its applied amount. Thus, the question remains whether PTX also contributes to the pharmacodynamic effect of ethanol. Other cases of paclitaxel poisoning have not been described. However, the literature describes poisoning by *Taxus* plants, in which paclitaxel itself is present. 

Labossiere et al. carried out an overview of such poisonings in order to map clinical cases of acute yew intoxication and the treatment strategy, finding that the majority of unintentional poisonings were not life-threatening, and for life-threatening intoxications, different treatment strategies were chosen in an effort to target the specifics of this intoxication. These treatment strategies had little demonstrable effect. The documented mortality of intentional intoxications in this study was 42% [20]. After ingestion, taxines (Figure 1) are very quickly absorbed into the bloodstream, also due to the low pH in the stomach, and cause nausea, vomiting, dizziness and abdominal pain 150]. The first symptoms after ingestion begin to appear in 30 min to 1.5 h, while death usually occurs between 1.5 and 24 h [149]. At this time, the concentrations of substances have a toxic effect on the organism. In addition, toxicological analysis is necessary for the later determination of the diagnosis. Dahlqvist et al. observed approximately 30 min after hospital admission that taxine B concentrations decreased from approximately 125 ng/mL (before haemodialysis) to 80 ng/mL 2 h after dialysis [150]. Another clinical case of suicide attempt, when poisoning was analytically proven, describes the concentrations found in the yew. Concentrations (ng/mL) were measured in serum and urine on the day of admission and two days after treatment for PTX (serum 0.52 and <0.016, urine 0.72 and 39.6), cephalomannine (serum <0.012 and <0.012, urine 1.46 and 166.2) and for Baccatine III (serum 2.95 and 0.32, urine 51.6 and 105.3) [151]. Clinical studies of patients who have survived intoxication have distorted laboratory results due to the intensive initial therapy. However, it can be assumed that the concentrations mentioned above could be even higher before the therapy. Therapy of acute symptomatic intoxication and cardiopulmonary resuscitation of patients includes advanced life support, according to the European Resuscitation Council (ERC) guidelines 2021—securing airways, ventilation, circulation by administration of crystalloid solutions, catecholamines, calcium, administration of antidote for digoxin intoxication for its described possible positive effect on taxane intoxication, administration of activated charcoal to the GIT via a nasogastric tube to reduce further resorption of toxins, treatment of arrhythmias with amiodarone or lidocaine, magnesium, cardioversion—but their effectiveness is low [152,153]. Cardiopulmonary resuscitation is started immediately when electromechanical dissociation occurs, and if veno-arterial extracorporeal membrane oxygenation (VA ECMO) is available, it can be used for the therapy of a refractory shock. [151,154,155]. Due to the large volume of distribution, large molecular weight of taxins and high binding to proteins, the effectiveness of haemodialysis as an elimination method for the treatment of intoxication is unlikely [150], but it can be used to treat or stabilise the disruption of the acid–base and mineral balance [20]. 

The mechanism of the acute toxic effect on the organism is caused by the alkaloids taxine A, taxine B, isotaxine B, paclitaxel (taxol A), taxol B and taxicatin glycosite. Taxine A, but especially taxine B, binds to the sodium and calcium channels of cardiomyocytes and causes their blockade [150]. Blockage of sodium and calcium channels leads to impairment of the conduction system of the heart, reduction of excitability and conduction velocity, which causes the ineffectiveness of external cardiac stimulation or drug stimulation of the myocardium [154,156]. Taxines as antagonists of calcium channels of cardiomyocytes have a similar effect on the myocardium as antiarrhythmic drugs of group IV calcium blockers, while the cardioselectivity of taxines is higher than that of verapamil [157]. This mechanism of action also explains the very low effectiveness of administered Atropine in the treatment of bradycardia caused by intoxication and the low effectiveness of catecholamines for vasodilation [20]. The toxic effect is subsequently manifested by cardiovascular instability: a negative inotropic, chromotropic and dromotropic effect, at the same time, a highly proarrhythmogenic effect (disappearance of the P wave, AV blockade, prolongation of the QT interval, readiness of the myocardium for malignant rhythm disorder) and vasodilation up to vasoplegia [158]. Other manifestations of intoxication are nausea, vomiting, abdominal pain, disturbances of consciousness, and convulsions. Clinically, intoxication is manifested by impaired consciousness, bradycardia (especially at the beginning), hypotension, cyanosis, and mydriasis, followed by bizarre tachyarrhythmias and dyspnoea, which can subsequently result in circulatory arrest and death. There is no causal therapy for intoxication [158]. 

### 4.2. Lethal Intoxications

Table 4 contains all (except one) published fatal plant intoxication by *Taxus*, including three previously unpublished cases. In total, it includes 38 fatal cases of *Taxus* poisoning, 19 males and 19 females. The average age of men and women was the same—29.7 years. Age ranged from 16 to 70 years, in both cases women. As for the usage form of the *Taxus* plant, in 27 cases, the needles of the plant alone were used, in 4 cases a decoction/tea from needles, in 5 cases needles in combination with a decoction, and in 1 case the bark was used. In 22 cases of fatal poisoning, it was suicide, in one case accidental intoxication, and in the remaining 15 cases, the manner of death was not stated. Medical history of psychiatric illness occurred in 14 cases, the most common being depression with suicidal thoughts. In 3 cases, no autopsy was performed, and in another 4 cases, no record of autopsy was given in the publication. Autopsy was performed in the remaining 31 cases.

Autopsy examination and findings in acute yew poisoning are mostly unremarkable. At the autopsy, only non-specific symptoms are detected—significant acute congestion of the internal organs, especially the lungs, liver, kidneys and brain, acute dilation of the heart cavities, and swelling of the brain and lungs, often with bleeding into the lung alveoli. Microscopic findings are also nonspecific. Thus far, no specific morphological autopsy or histological findings on body tissues and organs have been recorded in the professional literature, which would indicate or prove yew intoxication. An autopsy finding that points to possible yew intoxication is the evidence of yew plant parts—most often typical needles (whole, chopped, crushed) in the digestive tract (especially in the contents of the stomach, duodenum and small intestine). Thus, fatal yew intoxication can be detected by evidence of needle-like leaves or their fragments in the body (digestive tract) and toxicological analysis (evidence of taxanes), which is decisive for determining the cause of death. If needles or their fragments are not found in the body, or autopsy is not performable, the only evidence of yew intoxication remains a toxicological analysis to detect the presence of taxanes in the stomach contents or in the blood or other biological materials. The concentrations of these substances found in cases of fatal taxus poisoning depend on the amount, part (leaves, bark, berries) as well as the form (e.g., whole leaves, cut leaves, crushed leaves, leaf mash, decoction) of the ingested parts of the Taxus plant (Table 2). In the table containing 38 cases of fatal yew intoxication, toxicological analysis was performed in 23 cases (60.5%). The disadvantage was that not all of them carried out a quantitative analysis. The concentration range for taxine B (sometimes quantified together with isotaxine) in blood was from 150 to 80,900 ng/mL, in gastric contents from 2000 to 40,500 µg/mL. PTX concentrations in blood were less than 0.5 ng/mL, the concentration range for PTX in gastric contents was 20–82 ng/mL and in bile 24–800 ng/mL. The concentration range of 3,5-DMP (Figure 1) was in heart blood 31-820 ng/mL, in femoral blood 29–283 ng/mL, in urine 56–8700 ng/mL, in liver 161-918 ng/g, in the kidneys 275–418 ng/g, in the brain up to 35 ng/g, in bile 50–250 ng/mL, in the stomach contents 150–118,000 ng/mL, and in the duodenum 7800 ng/mL. As for other substances, the higher concentrations of 10-DAT (325–4900 ng/mL), BAC III (6.2–292 ng/mL), 10-DAB III (290–1690 ng/mL) and CEPH (37–482 ng/mL) found in bile were collected post mortem. The mentioned substances do not have a uniform distribution in the organism, but in order to determine the trend, it would be necessary to analyse the concentration in all cases of poisoning by the *Taxus* plant. Moreover, the concentrations can be helpful in the diagnosis of clinical poisonings or in post-mortem diagnosis.

The immediate cause of death is most often heart failure due to fatal cardiac arrhythmia. In all three cases (100%) of fatal yew intoxication published by us, pronounced mydriasis was found at autopsy. In the other 35 cases from the table, mydriasis was present in only five cases (14.3%). The observed differences can be explained either by not recording this clinical symptom, or by its absence in cases of fatal yew intoxication. Mydriasis, however, was observed in some clinical poisonings. In six cases from the table, the net weight of plant parts, needles found during dissection in the digestive tract, which ranged from 3.6 g to 200 g, is given. According to literature data, lethal oral doses of yew needles in humans are in the range of 0.6–1.3 g/kg of body weight [168]. If we assume that 1 g of yew leaves contains approximately 5 mg of taxines, the minimum toxic dose for humans is then calculated at 3.0–6.5 mg of taxines/kg of body weight. In one fatal case of yew intoxication (Case 1) described by us, 35 g of ground (crushed) and whole yew needles was found in the small intestine of a 64 kg man (for whom the lethal amount is 38.4–83.2 g of ingested yew leaves). This represents a fatal dose for the man, taking into account that the man vomited needles and a larger amount of needles was sucked out of his stomach at the hospital (about 1.5 h after being found). 

Thus far, serious intoxications due to the ingestion of drugs with Taxol as the active substance used for the treatment of cancer diseases have not been described in the scientific literature. On the other hand, poisonous yews are commonly found freely accessible in parks, gardens, orchards or cemeteries. This makes them potentially dangerous, and one must be vigilant, as even accidental ingestion of a *Taxus* part can lead to severe intoxication or even death. In addition, scientific publications describing the synergistic effect of paclitaxel due to co-administration with the extract of the *Taxus *spp. plant are beginning to appear in the literature. 

## 5. Conclusions and Future Perspectives

The most frequently used analytical methods were liquid chromatography with various detectors, and the highest concentration range for paclitaxel, BAC III, 10-DAP, 10-DAB III, 7-E-10-DAP and CEPH was observed in woody parts. The most studied plants were *T. cuspidata*, *T. chinesis* and *T. baccata.*

We summarized the pharmacokinetic (PK) parameters such as C_max_, T_max_, CL and AUC for different PTX formulations. However, there is still a need for more studies of PK parameters for these formulations in other ethnic groups and in geriatric patients as well as PK parameters for the newly developing co-administration of taxane extracts and PTX. Even despite premedication, there is still a predominantly organ-oriented toxicity during PTX administration, which could be better correlated not only with the dose but also with monitoring of the therapeutic level by fast and sensitive methods. At the same time, these would also be useful in the diagnosis of ever-present intoxications (often fatal) by *Taxus* spp. The already mentioned co-administration of PTX with *Taxus* spp. plant extracts brings the risk of intoxication, and for that reason, we have summarized the fatal intoxications with their medical history, clinical symptoms and autopsy findings, where apart from the characteristic mydriasis, no clear specific diagnostic sign was observed.

## Figures and Tables

**Figure 1 ijms-23-15619-f001:**
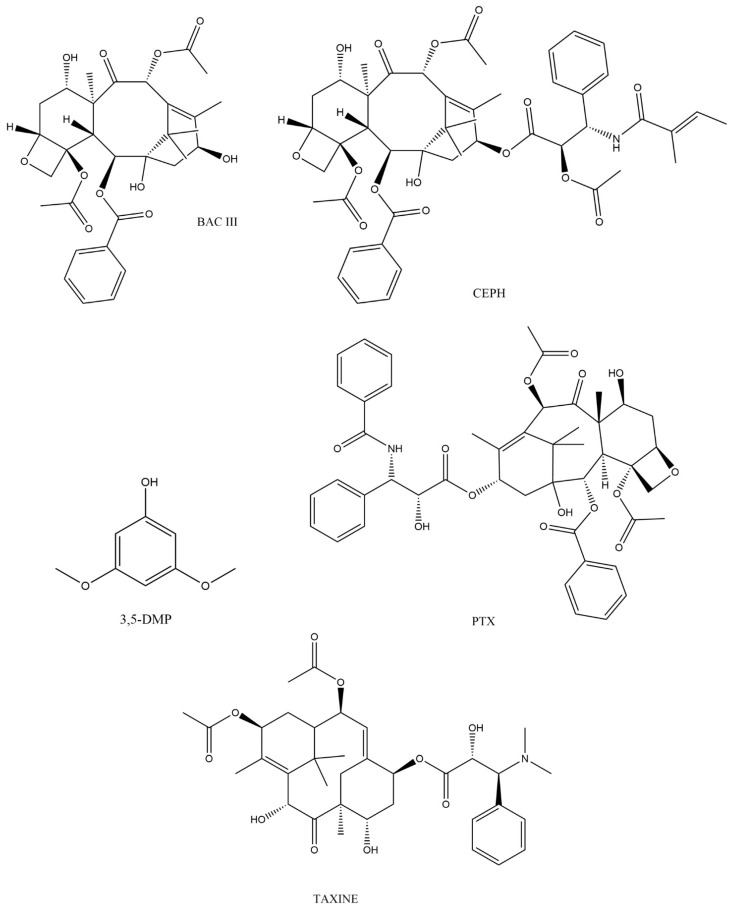
Structures of selected taxanes.

**Table 1 ijms-23-15619-t001:** Overview of the latest methods for analysing the composition of *Taxaceae* spp. for the last 5 years. The table does not include publications focused on in vitro, in vivo, clinical or forensic studies.

Taxus/Anatomical Parts	Analytes	Concentration of Analytes (µg/g)	Analysis Time (min)	Analytical Method	Citation
*T. Baccata* L. Red arils	Ascorbic acid Carotenoids Polyphenols Volatile compound profile Antioxidants Flavonoids	607–1450 33–54.2 12.2–538 na na 85–211	na 20–50 55	UV-Vis HPLC-DAD GC-MS	[28]
*T. chinesis* Bark, sapwood, and heartwood	Ethers Alcohols Acids Flavonoids Ketones Phenols Esters Aldehydes Pyridines	na na na na na na na na na	na 35	FTIR GC-MS	[29]
*T. chinesis* Stems	Fatty acids Phenols Volatile components Benzenes Acids Esters Ketones Alkanes	na na na na na na na na	60	GC-MS	[30]
*T. chinensis*, *T. cuspidata* *T. media*	*Amino acids* *Saccharides* *Acids* *Alcohols* *Amines* *Unspecified compounds*	na na na na na na	40.5	GC-MS	[31]
*T. chinensis* Leaf and wood essential oil hydrodistilled	Monoterpenes	Yields of the yellow oils 0.15 %, 0.11 % (*v*/*w*)	52	GC-MS	[32]
*T. baccata* Needles	Flavonoids	166,000–220,100	More than 100	LC–ESI–MS/MS	[33]
*T. chinensis* Heartwood (HW) Sapwood (SW)	Lipids Organic acids and their derivatives Nucleotides and their derivatives Flavonoids Amino acids and their derivatives Alkaloids Phenylpropanoids Vitamins Terpenes Carbohydrates	na na na na na na na na na na na	na	LC-MS	[34]
*T. baccata* L. Needles	10-DAB III	21.1–113.2	na	LC-MS/MS	[35]
*T. baccata* Needles	10-DAB III BAC III 10-DAT PTX CEPH	na na na na na	cca 10	UPLC-MS/MS LC-MS/MS	[36]
*T. baccata* Needles	PTX 10-DAB III 10-DAT BAC III CEPH	17.8–29.7 162.3–703.4 14.7–17.5 trace 19.2–29.3	na	UPLC-MS/MS	[37]
*T. baccata* L.	Flavonoids	204,260 ± 6020	cca 100	HPLC–DAD–ESI–MS/MS	[18]
*T. baccata* L. Red arils	Fatty acids Proteins Lipids Carbohydrates Amino acids 10-DAB III BAC III CEPH Taxinine M Taxol A Macroelements Microelements	7199–15,831 17,900–38,000 13,900–35,500 184,300–193,000 9167–15,831 3.9–19.8 2–6.3 0.05–0.18 0.02–0.13 0.02–0.10 8.6–8783.8 0.0013–25.37	More than 45 GC-FID: more than 175	LC-MS/MS GC-FID Macroelements, microelements, and trace metals ICP-OES	[38]
*T. cuspidata* *T. mairei* *T. media* Twigs	Secondary metabolites Amino acids Cofactors and vitamins Carbohydrates Lipids Nucleotides Energy-related metabolites Flavonoids PTX BAC III 10-DAT 10-DAB III	na na na na na na na *T. cuspidata* 1670 *T. mairei* 660 *T. media* 1220 *T. cuspidata* 800 *T. mairei* > 400 *T. media* > 200 *T. cuspidata* 800 *T. mairei* 200 *T. media* 800 *T. cuspidata* 800 *T. mairei* 1000 *T. media* 400	cca. 15 for untargeted analysis	HPLC-MS/MS UPLC-MS/MS	[23]
*T. media* *T. cuspidata* *T. mairei* *T. grandis* Twigs	Terpenoids Amino acids Flavonoids Phenylpropanoids PTX 7-epi 10-DAT 10-DAB III BAC III 10-DAT CEPH 7-E-DAB	na na na na *T. cuspidata* < 100 *T. media* 300 *T. mairei* < 100 *T. grandis* 0 *T. cuspidata* < 70 *T. media* 60 *T. mairei* < 10 *T. grandis* 0 *T. cuspidata* < 50 *T. media* 150 *T. mairei* 150 *T. grandis* 0 *T. cuspidata* 5 *T. media* < 20 *T. mairei* 20 *T. grandis* 0 *T. cuspidata* < 250 *T. media* > 300 *T. mairei* 100 *T. grandis* 0 *T. cuspidata* < 100 *T. media* < 160 *T. mairei* 20 *T. grandis* 0 *T. cuspidata* 300 *T. media* < 200 *T. mairei* 250 *T. grandis* 0	Untargeted metabolomic profiling: 15 Hormones: 8	LC-QTOF LC-MS/MS UPLC-MS/MS	[39]
*T. chinensis* *T. cuspidata* *T. media* Twigs and leaves	10-DAB III BAC III 7-xyl-10-DAT 10-DAT CEPH PTX 7-epi-PTX Flavonoids	*T. chinesis* 3–50 *T. cuspidata* cca 0–70 *T. media* 140–420 *T chinesis* 3.5–22.5 *T. cuspidata* 2–15 *T. media* 2–45 *T. chinesis* 420–1110 *T. cuspidata* 0–90 *T. media* >90 *T. chinesis* <72–430 *T. cuspidata* 144–288 *T. media* 216–288 *T. chinesis* <110–330 *T. cuspidata* 330–600 *T. media* 330–550 *T. chinesis* 15–90 *T. cuspidata* 60–95 *T media* <15–180 *T. chinesis* <1.5–3.7 *T. cuspidata* <3.7–7.4 *T. media* 1.5–22.2 *T. chinesis* 5400–10,800 *T. cuspidata* <5400–16,200 *T. media* 5400–10,800	5	UHPLC-MS/MS	[40]
*T. cuspidata* Branches and leaves	Polysaccharides	4.47 *w*/*w* %	na	Ion chromatograph UV-VIS	[41]
*T. fuana* *T. yunnanensis* Twig	Alkaloids Amino acids Hormones Lipids Terpenoids Phenylpropanoids Saccharides PTX 10-DAB III DAB III Flavonoids 10-DAT 7-epi-PTX 7-Epi 10-DAT CEPH	na na na na na na *T. Fuana* 27 *T. yunnanensis* 84 *T. fuana* 200 *T. yunnanensis* 500 *T. fuana* 10 *T. yunnanensis* 25 *T. fuana* 150,000 *T. yunnanensis* 115,000 *T. fuana* 15 *T. yunnanensis* 150 *T. fuana* max 10 *T. yunnanensis* max 15 *T. fuana* max 10 *T. yunnanensis* 250 *T. fuana* 50 *T. yunnanensis* 100	15	UPLC-MS/MS	[42]
*T. yunnanensis* Needles	PTX 10-DAB III	na na	15	HPLC-PDA	[43]
*T. cuspidata* Bark	Caffeic acid Chlorogenic acid Gallic acid p-Hydroxybenzoic acid Hydroxycaffeic acid Protocatechuic acid	30.5 ± 0.1 83 ± 2.2 20.4 ± 0.7 24.2 ± 1.6 239.8 ± 13 209.7 ± 5.6	na	HPLC-DAD	[44]
*T. cuspidata* Stems and leaves	10-DAB III BAC III 10-DAT CEPH PTX	212–354 20–56.6 51–91 36–79 46–92	30	HPLC-UV or DAD	[45]
*T. chinensis* Leaves	(E)-1-O-(p-coumaroyl)-3-methoxy-myo-inositol p-Hydroxybenzaldehyde p-Hydroxybenzoic acid Palmitic acid Protocatechuic acid Sciadopitysin Ginkgetin Sequoiaflavonoid Taxacine 10-DAB III 5-deacetyltaxachitriene B Makisterone C 7-β-xylosyl-10-decetyltaxol Taxiphyllin	na na na na na na na na na na na na na na na	na	NMR HR-MS TLC	[46]
*T. baccata* L. *T. media* Arils	Carotenoids Xanthophyll esters	*T. media* 20.33–58.78 *T. baccata* 17–19.55	na	HPLC-DAD-ESI/APCI-MS*^n^*	[47]
*T. baccata* L. Red arils	Taxol A 10-DAB III BAC III CEPH Taxinine M 3,5-DMP	0.07–5.5 5.6–38 1–28 0.04–7.2 0.04–2 0.04–0.82	15.5	HPLC-MS/MS	[48]
*T. cuspidata* Needles	PTX 10-DAB III BAC III DXT DOT 7-EPT CEPH 10-DAT	na na na na na na na na	40	HPLC	[49]
*T. wallichiana* Needles	Total phenolic content Total flavanol content Total flavonoid content Total tannin content	92,670 ± 680 na 84,660 ± 520 na	na	HPLC-PDA	[50]
*T. Wallichiana* Leaf, stem, bark and roots	Docetaxel PTX	na 9.31–36.46	35 20	HPLC-UV QToF/MS	[51]
*T. chinensis* Needles	10-DAB III BAC III 7-xyl-10-DAT 10-DAT CEPH 7-epi-10-DAT PTX	434.75–626.44 45.51–120.48 351.44–546.95 104.05–236.49 42.74–105.69 94.45–166.44 85.79–162.75	25	HPLC	[52]
T. media T. mairei Leaves	Flavonoid content	*T. media* 14,464 *T. mairei* 19,953	15	UPLC-ESI-MS/MS	[53]

na—not applicable.

**Table 2 ijms-23-15619-t002:** Characteristics of selected taxanes, including molecular weight, general formula, polarity. Data were obtained from https://pubchem.ncbi.nlm.nih.gov/ (accessed on 5 November 2022).

Compound	Molecular Formula	Mw (g/mol)	Topological Polar Surface Area
Paclitaxel	C_47_H_51_NO_14_	853.98	221 Å^2^
Cephalomannine	C_45_H_53_NO_14_	831.9	221 Å^2^
Baccatine III	C_31_H_38_O_11_	586.6	166 Å^2^
Brevifoliol	C_31_H_40_O_9_	556.6	140 Å^2^
9-Dihydro-13-acetylbaccatine III	C_33_H_42_O_12_	630.7	175 Å^2^
Taxine A	C_35_H_47_NO_10_	641.7	160 Å^2^
2-Deacetyltaxine A	C_33_H_45_NO_9_	599.7	154 Å^2^
10-Deacetylbaccatine III	C_29_H_36_O_10_	544.6	160 Å^2^
Taxine B	C_33_H_45_NO_8_	583.7	134 Å^2^
Isotaxine B	C_33_H_45_NO_8_	583.7	134 Å^2^
Taxinine M	C_35_H_42_O_14_	686.7	na
10-Deacetyltaxol	C_45_H_49_NO_13_	811.9	215 Å^2^
7-Epitaxol	C_47_H_51_NO_14_	853.9	221 Å^2^
7-Epi-10-deacetylbaccatine III	C_29_H_36_O_10_	544.6	160 Å^2^

na—not applicable.

**Table 4 ijms-23-15619-t004:** Fatal cases of poisoning by *Taxus* spp.

Anamnesis, Case History Clinical Findings	Autopsy Findings	Taxane Concentration	Analytical Method	Cause and Manner of Death	Ref.
**20-year-old female,** prepared tea (decoction) from three spoons of yew needles, let it brew for 0.5 h, then drank the tea and ate the infused yew needles with bread. After 1 h dizziness, strong palpitations, ringing in the ears, burning in the whole body, vomiting (vomits contained needles), soon unconsciousness. On admission to the hospital, pallor, tachycardia, hypotension, later hypertension, then tachycardia again. 24 h after ingestion death. Medical history: psychological problems, suicidal thoughts	Isolated needles of Taxus leaves found in the stomach throughout the small intestine and a large number of Taxus needles Mild necrosis of the gastric mucosa with small haemorrhages Dilation of heart chambers Blood flow to internal organs Histology: myocardium interstitial oedema, signs of fatty degeneration in the liver and kidneys	na	na	**Cause of death:** Heart failure in taxus intoxication **Manner of death:** Suicide	[159]
**24-year-old male,** a year after the death of his ex-fiancée, he suffered from depression, often expressing suicidal thoughts. He went to a ball, drank very little alcohol. When he was returning home, his current fiancée told him that she was probably expecting his child. About 4 h later, he was found at home lying in bed unconscious, breathing. On admission to the hospital, deep unconsciousness, gasping breathing, pale skin, cyanosis of the face, neck and upper chest, very wide unresponsive pupils (mydriasis), weak pulse, weak irregular heart activity, poor reflexes, unmeasurable blood pressure. Despite immediate treatment, respiratory and cardiac arrest in a short time → death.	In the stomach, approx. 200 mL of porridge mixed with vegetable matter (39 g of vegetable matter in total). In the duodenum, only a moderate amount of food (mass) mixed with isolated plant components. Slightly greenish mucus in the larynx, acute enlargement of the heart chambers, acute congestion and pulmonary oedema, severe acute congestion of the spleen, severe acute congestion of the liver, congestion and swelling of the kidneys, severe swelling of the brain Microscopic examination of stomach contents: found cut-up *Taxus baccata* presses	Taxine	UV Thin-layer chromatography	**Cause of death:** acute circulatory failure in case of poisoning by *T. baccata* **Manner of death:** Not reported	[160]
**28-year-old female,** teacher, she knew the toxicity of yew, she ingested 4 to 5 handfuls of yew needles with suicidal intent. A yew tree was in the garden in front of her house. About 1 h after taking dizziness, nausea, no vomiting, diffuse abdominal pain. Taken to the hospital, in a drowsy state, tonic-clonic seizure in the reception room, then unconsciousness, tachycardia, wide pupils (mydriasis), weak, later no reflexes, respiratory arrest, unmeasurable blood pressure. Subsequently, intubation, artificial ventilation, intensive care. In a few minutes, asystole → followed by cardiac stimulation and resuscitation, after 50 min resuscitation ended, heart, circulation and breathing stopped → death. Performed **gastric lavage** with 10 litres of water, a large number of needles and food residues obtained during the lavage.	2 to 3 cm long coniferous yew twigs found in the stomach Acute haemorrhagic gastroenteritis Acute congestion of the liver, kidneys and spleen, pulmonary and myocardial oedema, punctate subpleural haemorrhages, flaccid and dilated all cavities of the heart Histology: small necrosis of cardiomyocytes of the left ventricle of the heart, acute small hepatocellular necrosis in the liver, incipient diffuse fatty degeneration of the liver	na	na	**Cause of death:** Heart failure **Manner of death:** Suicide	[161]
**40-year-old female,** she intentionally ate about 150 yew leaves (*Taxus baccata*). 2 h after taking it, she sought medical help due to vomiting and abdominal pain. Admitted to hospital. Symptoms: hypotension, followed by shock and respiratory arrest. Immediate resuscitation (external cardiac massage, intubation, artificial ventilation, heart rhythm disorder on EKG, right ventricular pacemaker inserted). Aspiration of gastric fluid performed immediately after intubation showed the presence of yew leaves (needles). After 3 h, ventricular fibrillation unresponsive to treatment, death 5 h after yew leaves ingestion. Medical history: chronic psychosis.	Autopsy not performed	na	na	**Cause of death:** Cardiogenic shock **Manner of death:** Suicide	[162]
**22-year-old male**, 4th year university student of agriculture. Found on a winter morning lying on the “step” of a high bridge; he did not respond. A bag with washed clothes was next to the body. He was seen walking down the road a few minutes before. After an examination, the doctor at the emergency room declared the man dead; alcohol test negative. Medical history: loner, **marijuana** user. A bag of marijuana was found in his pocket.	Fresh green grass-like leaves found in jejunum—identified by agronomist as yew leaves Smell of fresh yew leaves detected during autopsy Cannabinoids present in urine at a concentration of more than 75 ng/mL	na	GC-FID	**Cause of death:** Yew poisoning **Manner of death:** Not reported	[163]
**19-year-old female**, found dead. She had told her friend she was considering suicide.	Green particles found in the stomach, identified as parts of *Taxus baccata* Congestion of lungs, liver, kidneys Dilation (expansion) of heart chambers	na	na	**Cause of death:** Yew poisoning **Manner of death:** Suicide	[164]
**70-year-old female,** admitted to the hospital after a suicide attempt by diazepam intoxication. After recovery, she was taken to a psychiatric hospital, where she was allowed to walk around the campus. After several days of severe abdominal pain, she admitted to the doctor that she had eaten parts of Taxus tree bark. Hypotension, bradycardia, cardiac arrest → resuscitation unsuccessful → death.	Autopsy not performed	na	na	**Cause and manner of death:** Not reported	[164]
**23-year-old female**, hospitalized at psychiatry, found dead.	Green mass with plant fibres found in the stomach and duodenum—identified as parts of Taxus leaves. Congestion of organs Dilated heart chambers	na	na	**Cause of death:** Yew poisoning (intoxication) **Manner of death:** Not reported	[164]
**26-year-old female**, psychiatric patient, found dead. She was known to hear voices urging her to commit suicide.	The stomach and duodenum contained a brown liquid in which green plant parts were present. These were identified as *Taxus baccata* leaf fragments. Congestion of organs Dilated heart chambers	na	na	**Cause of death:** Yew poisoning (intoxication) **Manner of death:** Not reported	[164]
**37-year-old female**, prisoner, found dead in her prison cell bed. She was considering suicide. A small *Taxus* plant was found in her prison cell.	Green plant parts were found in the stomach and duodenum, which were identified as leaves and leaf parts of *Taxus baccata*	na	na	**Cause of death:** Yew poisoning (intoxication) **Manner of death:** Suicide	[164]
**19-year-old male**, found dead by friends in a remote cellar after leaving home two days earlier. He was lying on the couch, partially undressed. An empty teapot with fragments of brown-green leaves on the sides was found on the floor. Boiled and pressed leaves with a teaspoon were piled on the carpet near the teapot. Cellar was with no signs of vomiting or diarrhoea. Medical history: depression	Fragments of greenish needle-like leaves found in the mouth, oesophagus, stomach and intestines (not in the anus) (30% of the stomach contents were made up of leaves, i.e., 150 out of 500 g) All organs were heavily blood-stained, bronchial epithelium was inflamed Histology: significant to severe congestion of organs, slight damage to brain neurons, massive desquamation of the alveolar epithelium in the lungs, small to medium vacuolar degeneration of the myocardium	3,5-DMP stomach 20,000 ng/g cardiac blood 0.32 mg/kg, 1.31 g/kg ethanol	HPLC, UV, GC-MS, IR ^1^H-NMR	**Cause of death:** Acute cardio-circulatory failure **Manner of death:** Not reported	[165]
**43-year-old male**, schizophrenic, undergoing treatment several times, attempted suicide by cutting his wrist in the past. He was prescribed antipsychotic perazine (Taxilan) for a long time. Due to frequent side effects, he was looking for an alternative medicine. For this purpose, he bought a small yew tree, made a decoction from its leaves and drank it. He later told the nurse that he was tolerating yew better than perazine, and that he intended to replace this antipsychotic drug with regular use of yew tea. It is not known when he drank the second yew tea. Then, he sat with the nurse, complaining of nausea and lack of blood circulation. He told her that he had taken *Taxus baccata* leaves but did not specify the amount. He started vomiting, the nurse carried him to bed. When she went to check on him after 3 h, she found him lying dead in bed. There was an empty tea strainer in the kitchen, which the nurse later cleaned up.	Acute organ congestion, massive brain oedema, haemorrhagic pulmonary oedema, stomach dilatation Histology: significant interstitial oedema and signs of myocardial hypoxia, mild fatty degeneration of the liver, massive dilatation of submucosal gastric vessels Smears of aqueous stomach contents were not successful in identifying particles typical for *Taxus baccata*	11,000 ng taxine/g blood (sum of taxine B and isotaxine B)	LC-MS LC-MS/MS	**Cause of death:***Taxus baccata* intoxication **Manner of death:** Not reported	[166]
**24-year-old male,** ingested yew needles with the aim of committing suicide.	Many yew leaves present in the stomach; other autopsy findings not stated	Taxine B/isotaxine B Blood-105 ng/g Stomach content 2000 ng/g Urine 0 ng/mL	LC-MS/MS	**Cause of death:** Not reported **Manner of death:** Suicide	[167]
**33-year-old female,** a glass with red yew fruits was found near the dead body. Yew plant material was ingested; it is not known whether she ate any berries or drank a decoction of them.	No plant material was found in the body; other autopsy findings not stated	Taxine B/isotaxine B Blood-174 ng/g Stomach content 50,000 ng/g Urine 3000 ng/mL	LC-MS/MS	**Cause of death:** Not reported **Manner of death:** Not reported	[167]
**23-year-old female,** long-term psychiatric treatment, found dead in her apartment. Information materials about toxic plants found in the apartment. Previous suicide attempts. The police investigation found that the woman sought medical help 2 weeks before her death due to symptoms similar to yew poisoning—dizziness and arrhythmia. However, she refused hospitalisation.	Approx.. 200 g of greenish-brown plant particles, identified as whole leaves and fragments of *Taxus baccata* leaves, were found in the duodenum and large intestine Advanced decomposition of the body	Cardiac blood (ng/mL) 47 Femoral blood (ng/mL) na Urine (ng/mL) 8700 Brain (ng/g) < 30 Liver (ng/g) 161 Kidney (ng/g) 275 duodenum (ng/g) 7800	HPLC-PDA HPLC-UV	**Cause of death:** Taxus intoxication (poisoning) **Manner of death:** Suicide	[168]
**20-year-old male,** found dead in the park inappropriately dressed for the season.	Approx. 150 g of green leaves found in the stomach and duodenum, identified as parts of *T. baccata* Significantly dilated pupils (mydriasis), blood flow to the lungs and brain, dilated heart chambers	Cardiac blood (ng/mL) 97 Femoral blood (ng/mL) 29 Urine na Brain (ng/g) 35 Liver (ng/g) 512 Kidney (ng/g) 382 Stomach content (ng/g) 13,400	HPLC-PDA HPLC-UV	**Cause of death:** Taxus intoxication (poisoning) **Manner of death:** Not reported	[168]
**26-year-old male,** found dead in an upstairs room of the house, lying dressed on the bed, his face resting on a pillow. In the room found a blender containing the remains of green porridge and a bowl with green porridge stuck to it.	Greenish plant material was found in the stomach, in which fragments of yew needles were identified Marked swelling of the brain, acute organ congestion	Cardiac blood (ng/mL) 528 Femoral blood na Urine na Brain na Liver (ng/g) 918 Kidney (ng/g) 418 Stomach content (ng/g) 118,000	HPLC-PDA HPLC-UV	**Cause of death:** Taxus intoxication (poisoning) **Manner of death:** Not reported	[168]
**23-year-old male**, student, found dead on the stairs of the convention centre around lunchtime Before his death, he felt sick, dizzy. He was last seen drinking tea 2 h before his death. The owner of the apartment found a small plastic bag with yew leaves in his backpack	200 mL of stomach contents contained green particles Perfusion of organs (brain, liver, spleen, kidneys)	Cardiac blood (ng/mL) 110 Femoral blood (ng/mL) 217 Urine na Brain na Bile (ng/g) 175 Kidney (ng/g) na Stomach content (ng/g) 1400	HPLC-PDA HPLC-UV	**Cause of death:** Yew intoxication (poisoning) **Manner of death:** Not reported	[168]
**16-year-old female**, found dead in the bathroom of her parents’ apartment. Several hours before death registered dizziness, nausea, abdominal pain, unconsciousness. Medical history: mental disorders, depression	A large number of fragments of *Taxus baccata* leaves found in the mouth, oesophagus, stomach and duodenum, as well as in the trachea, wide dilated pupils (mydriasis), pulmonary oedema, acute congestion of organs (liver, spleen and kidneys)	Cardiac blood (ng/mL) 31 Femoral blood na Urine (ng/mL) 2700 Brain na Liver na Kidney na Stomach content (ng/g) 600	HPLC-PDA HPLC-UV	**Cause of death:** *Taxus baccata* poisoning **Manner of death:** Suicide	[168]
**41-year-old male**, found dead lying on the ground near the parking lot where his car was parked. His hands were clenched in spasms. History and cause of death unknown, suspected epileptic seizure	In the stomach and small intestine (not in the large intestine) found fragments of greenish needle-like leaves (chopped leaves) identified as yew. Autopsy and histological findings in agreement with the literature Examined bone marrow—oedematous, slightly hypocellular, with irregular distribution of hematopoietic cells	3,5-dimethoxyphenol quality	GC-MS	**Cause of death:** Yew poisoning (intoxication) **Manner of death:** Not reported	[169]
**30-year-old male**; found dead in his bed; near the bed found a light brown vegetable matter in a plastic bag—it looked the same as dried or partially macerated coniferous leaves	Plant material found in the stomach similar to that found in the plastic bag near the bed—botanically identified as *Taxus baccata* leaf fragments All organs were markedly congested Non-specific morphological findings	3,5-DMP Blood 146 ng/mL Urine 56 ng/mL Bile 50 ng/mL Gastric content 360 ng/g	GC-MS	**Cause of death:** Fatal cardiac arrythmia **Manner of death:** Not reported	[170]
**28-year-old male**, found dead in the basement of the family home. He ingested a decoction of yew mixed with sodium hydroxide	Acute catarrhal inflammation of the oesophagus with mucosal sloughing, acute superficial gastritis crushed plant material in the stomach and duodenum—needles and small twigs of *T. baccata* (3.6 g in dry state), blood effusions under the pleura, epicardium and in the soft coverings of the skull, brain swelling, haemorrhagic pulmonary oedema, congestion of internal organs, liquid blood	The presence of 3,5-dimethoxyphenol and 11-nor-A9-tetrahy-drocanabi-nol-carboxylic acid was detected in blood and urine. The presence of 3,5-dimethoxyphenol was further demonstrated in gastric and duodenal contents. In the gastric contents and urine, substances of the same nature as those contained in yew were detected by TLC.	TLC	**Cause of death:** *T. baccata* poisoning **Manner of death:** Suicide	[171]
**20-year-old male**, found dead in the area of the psychiatric hospital	Presence of a large number of green needles in the stomach, less in the small intestine—identified as yew Dark red-purple post-mortem spots, (dilated pupils)—mydriasis, conjunctival congestion signs of suffocation with dilatation of the right ventricle of the heart Histology: myocardial interstitial oedema, rarely increased eosinophilia of cardiomyocytes	The presence of taxine B, isotaxine B and other yew substances was proven in the blood and stomach contents. The presence of 3,5-dimethoxyphenol was proven in the blood and stomach contents.	LC/MS GC/MS	**Cause of death:** *Taxus baccata* poisoning **Manner of death:** Suicide	[171]
**43-year-old female**, intentionally ingested an unknown amount of Taxus needles. Dizziness, impaired consciousness, dysrhythmias, circulatory failure, asystole, death.	Not reported	na	na	**Cause of death:** Not reported **Manner of death:** Suicide	[158]
**22-year-old male**, a professional gardener with a history of drug abuse (cannabis), brought to the hospital for detoxification. After 3 weeks, he was transferred to another hospital, where he announced his intention to commit suicide using poisonous plants. In the following days, he collected twigs from the hospital garden (which were later identified as yew twigs) and brought them to his room. He ingested an unknown amount of yew leaves. On the morning of the day of death, severe nausea, heaviness without vomiting, hypotension. The evening shortly before his death, he was found unresponsive in bed, with breathing difficulties.	Vegetables and small green needle-like particles found in the stomach Flat green parts of plant materials found on the tongue and in the oesophagus—needle-like yew leaves Congestion and cyanosis of internal organs Brain and lung oedema Small haemorrhages in the epicardium Expansion (dilatation) of the atria and the right heart ventricle	**PTX** Stomach content 20 ng/mL Urine < 0.5 ng/mL Cardiac blood < 0.5 ng/mL Femoral blood < 0.5 ng/mL Bile 24 ng/mL Brain < 0.5 ng/g **10-DAT** Stomach content 36 ng/mL Urine < 0.5 ng/mL Cardiac blood < 0.5 ng/mL Femoral blood < 0.5 ng/mL Bile 4900 ng/mL Brain < 0.5 ng/g **BAC III** Stomach content 4.5 ng/mL Urine 19 ng/mL Cardiac blood < 0.5 ng/mL Femoral blood < 0.5 ng/mL Bile 6.2 ng/mL Brain < 0.5 ng/g **10-DAB III** Stomach content 132 ng/mL Urine 200 ng/mL Cardiac blood 12 ng/mL Femoral blood 7.3 ng/mL Bile 290 ng/mL Brain < 0.5 ng/g **CEPH** Stomach content 23 ng/mL Urine 1 ng/mL Cardiac blood < 0.5 ng/mL Femoral blood < 0.5 ng/mL Bile 37 ng/mL Brain < 0.5 ng/g **3,5-DMP** Stomach content 150 ng/mL Urine 7250 ng/mL Cardiac blood 110 ng/mL Femoral blood 60 ng/mL Bile 250 ng/mL Brain < 2 ng/g	HPLC-MS LC-MS/MS	**Cause of death:** Circulatory arrest **Manner of death:** Suicide	[172]
**38-year-old male**, found dead in bed. Small fragments of greenish needle-like leaves found at the site (on the table in the bowl, next to the bed, on the blanket, on the chin and mouth of the deceased man). Scissors on the table, which he probably used to cut the leaves into small pieces before eating. According to the police report, the green leaves were identified as Christmas tree needles.	Fragments of green needle-like leaves found on the chin, mouth and stomach Non-specific autopsy findings	**PTX** Stomach content 82 ng/mL Bile 800 ng/mL Urine, cardiac blood, femoral blood < 0.1 ng/mL **10-DAT** Stomach content 39.1 ng/mL Bile 325 ng/mL Urine, cardiac blood, femoral blood **BAC III** Stomach content 14.7 ng/mL Bile 292 ng/mL Urine 64.5 ng/mL Cardiac blood 8.25, femoral blood 8.5 ng/mL **10-DAB III** Stomach content 69.5 ng/mL Bile 1690 ng/mL Urine 74 ng/mL Cardiac blood 19.9 ng/mL Femoral blood 19.8 ng/mL **CEPH** Stomach content 45.7 ng/mL Bile 482 ng/mL Urine, cardiac blood, femoral blood, <0.1 ng/mL **3,5-DMP** Stomach content 423 ng/mL Bile 138 ng/mL Urine 5750 ng/mL Cardiac blood 820 ng/mL Femoral blood 283 ng/mL		**Cause of death:** Taxine intoxication **Manner of death:** Suicide	[173]
**43-year-old male**, for suicidal reasons he ate the leaves of the common yew (*T. baccata*). Symptoms: severe hypokalaemia, ventricular arrhythmia, hemodynamic instability, respiratory insufficiency, acid-base imbalance disorder, hepatic dysfunction, renal failure, coma, and 12 h after ingestion of *T. baccata*, despite intensive medical care and resuscitation, death. Stomach lavage performed in the hospital with evacuation of *T. baccata* leaves	Congestion and pulmonary oedema Brain oedema Enlarged liver Histology: acute dystrophic changes in the liver; dilation of blood vessels in the myocardium	na	na	**Cause of death:** Not reported **Manner of death:** Suicide	[174]
**43-year-old male,** found unconscious at home by his family. Taken to hospital: coma (GCS 3 points), tachycardia, acute respiratory and hepatic failure, hypokalaemia, hypotension, mixed acidosis. Gastric lavage, administered activated charcoal, supportive hemodynamic treatment. After 4 h of admission, unresponsive cardiac arrest → death. Medical history: about 10 years of repeated depressive disorders treated irregularly, recently without treatment, he gradually stopped working, became interested in computers and suicide methods, and consulted information on the Internet about the yew plant and its toxic effects. A farewell letter and an empty plastic bag with the remains of green leaves were found near the body. A police investigation revealed that several yew bushes were found near the house.	In the stomach approx. 450 mL of dark liquid—partially digested food fragments and green-brown plant particles—botanically identified as whole leaves and fragments of yew leaves Non-specific symptoms of intoxication: swelling of the brain, blood flow to internal organs, pleural, pericardial and peritoneal effusions, hepatomegaly (2600 g) Histology: dystrophic changes of the liver and kidneys	na	na	**Cause of death:** Heart failure Fatal cardiac arrhythmia **Manner of death:** Suicide	[175]
**25-year-old male**, drank a decoction of red yew with suicidal intent. Shortly after consuming the concoction, he reported it to his girlfriend, who called for medical help. When admitted to the hospital, about 1 h after taking it, conscious, with stable blood pressure, sinus tachycardia. Gastric lavage in the emergency room and in the hospital ward. Acidosis, liver and kidney functions and mineralogram were normal. After about 0.5 h ventricular fibrillation, cardiogenic shock, unconsciousness. Intubation, artificial pulmonary ventilation, repeated cardiopulmonary resuscitation, established temporary cardiostimulation. Hypokalaemia, acidosis, haemodialysis treatment without effect, cardiogenic shock, asystole → death approx. 6 h after administration.	Yew needles found in the small intestine. cerebral blood flow Congestion of the lungs with haemorrhages in the pulmonary alveoli Dilatation (enlargement) of the right atrium and heart chamber	na	na	**Cause of death:** Cardiac arrest due to red yew intoxication **Manner of death:** Suicide	[176]
**22-year-old female**, after leaving the exhibition in the botanical garden she collapsed, tonic-clonic convulsions, no pulse, not breathing (apnoea). Laic cardiopulmonary resuscitation was started, medical help was called, transport to the hospital. Symptoms: hypotension, bradycardia, restoration of circulation, plant material found in the mouth, green plant material aspirated from the stomach with a nasogastric tube; a bag of plant material, identified as *T. baccata* was found in the woman’s purse. Despite the intensive care of heart rhythm disorders, after approx. 1.5 h death since admission to hospital.	Not reported	Serum 3,5-DMP 86.9 ng/mL Gastric content 73,200 ng/mL taxine B serum 80,900 ng/mL Gastric content 40.5 mg/mL	HPLC-MS	**Cause of death:** Not reported **Manner of death:** Not reported .	[177]
**19-year-old female**, admitted to a closed department of a psychiatric hospital due to a suicide attempt. Found dead in bed in the morning. A document about the toxic effects of yew was found in her notebook. Inspection of the apartment—found several yew needles, 1 litre of tea from yew leaves, a teacup. No traces of yew leaves were found on the body, mouth and pharynx.	Autopsy not performed.	BAC III 10-DAB III 10-DAT Taxine B Isotaxine B PTX CEPH 3,5 DMP	LC-MS LC-MS/MS GC-MS HPLC-PDA	**Cause of death:** Not reported **Manner of death**: Based on the document in her notebook, yew leaves and a half-empty teacup, it was concluded that it was suicide by means of yew tea.	[22]
**30-year-old female,** found dead at home lying in bed. Green plant material in a plastic bag was found near the bed.	A mass of dark green needle-like leaves found in the stomach, similar to those found in the plastic bag—identified as *Taxus baccata* leaves Acute congestion of the organs, significant pulmonary oedema	3,5-DMP from Blood Kidney Bile Brain	GC-MS	**Cause of death:** Yew intoxication **Manner of death:** Suicide	[149]
**40-year-old female,** had a video call with a friend, she suddenly developed breathing difficulty. The friend suggested calling an ambulance, the woman was unable to respond, the friend called her mother, who called an ambulance. Rescuers tried resuscitation, but without success. Numerous branches of plant material found at the scene of death, as well found in the stomach at autopsy (as if she had cooked/consumed it). Medical history: during her life, she had a tendency to collect natural herbs and edible plant material from forests.	Numerous partially digested plant leaves in the shape of needles and small twigs found in the stomach contents Significant pulmonary oedema Hepatosplenomegaly Obesity (BMI 38.8 kg/m^2^) Hepatic steatosis	MAT 1 MAT 2 Taxine B Isotaxine B MHDAT1 MHDAT 2 TAT MHTAT Blood	LC-MS/MS	**Cause of death:** Acute cardiorespiratory arrest. **Manner of death:** Not reported	[178]
**20-year-old female**, brought to the emergency department by a friend. She stated that after reading information on the Internet, she consumed a large amount of yew leaves as a suicide attempt. Symptoms: 4 h after ingestion, nausea, flushing, hypotension, tachycardia, generalized tonic-clonic seizure, cardiac arrest. After resuscitation, restoration of spontaneous circulation, intubation, then arrhythmia. A small amount of green fluid and leaves present in the gastric lavage. Activated charcoal was administered, a transvenous pacemaker was introduced for heart rhythm disturbances, and she was transferred to the coronary unit, subsequently arrhythmias—tachycardia unresponsive to anti-arrhythmics. After consultation with the family, invasive medical measures were stopped and resuscitation ended—death shortly thereafter. Medical history: anxiety, bipolar disorder, obsessive-compulsive disorder, self-harm since age 12.	Not reported	na	na	**Cause of death:** Not reported **Manner of death:** Suicide	[179]
**35-year-old male**, called 911 himself because he felt sick—his head was spinning. When medical help arrived, he was found in the car in the driver’s seat, with slightly impaired consciousness (somnolence), vomited. He stated that he had ingested ground needles mixed into yogurt. He repeatedly passed out on the spot, had apnoeic phases and convulsions, vomited contents with ground green needles, aggressive, resisted examination Transferred to the hospital for emergency admission—impaired consciousness, bradycardia, asystolic pauses, unmeasurable blood pressure and pulse, post-defecation, vomiting, acral cyanosis, pronounced mydriasis, vomiting and full of ground needles in the mouth. Resuscitation started—intubation, inserted nasogastric tube, gastric lavage—brown content with ground needles came out, external cardiostimulation, blood and urine collection for toxicological examination. Despite 185 min of resuscitation on EKG asystole, death was declared. Toxicology results: traces of 3,5-dimethoxyphenol (GC-MS)—a yew metabolite—were proven in the urine sample. In the car, on the back seat 2 grinders containing ground (crushed) green yew needles, on the driver’s seat and in front of the seat on the floor ground (crushed) green plant—yew, in the vehicle two plastic bags filled with green needles and yew twigs. Next to the car, a sweatshirt soiled by a green plant. Medical history: according to his brother, he was hospitalized in psychiatry a year earlier, ID no. without treatment.	A green mushy mass with flat green needles present in the small intestine—approx. 35 g in total a coating of black liquid on the tongue, in the pharynx and in the oesophagus Thin-mushy black matter in the stomach and duodenum Congestion of the gastric mucosa Pronounced mydriasis Significant blood flow to internal organs. Liquid blood Enlarged right atrium and heart chamber Marked swelling of the brain Swelling of the lungs Petechiae under the pleura Slightly underweight (64 kg) Signs after resuscitation Histology: significant swelling of the brain and dystrophic changes of nerve cells, swelling and acute emphysema of the lungs, dispersed acute ischemic changes in the myocardium with signs of acute blood circulation failure, dystrophic changes up to necrosis of individual liver cells; dystrophic changes in kidney tubule cells	**Urine:** qualitatively 3,5-dimetoxyphenol —a yew metabolite Caffeine Nicotine	GC/MS	**Cause of death:** Cardiac failure due to acute intoxication with *T. baccata* **Manner of death:** Most likely accidental intoxication	Our case I 2022
**24-year-old female**, found in the afternoon lying dead in the hallway of the rented apartment, in front of the toilet. A jug filled with coniferous branches, with a cooking pot and a strainer, was found in the kitchen. The previous evening, she wrote an e-mail to her friend with the text “I made mistakes, I love you”. In her notebook, the keywords “Suicide” and “Yew” were found in the history of the Internet browser.	Significant mydriasis (0.7 cm) Congestion of internal organs Swelling of the brain and lungs	**Kidney:** qualitatively present taxine **Tea from the place:** qualitative taxine **Yew needles:** qualitative taxine	TLC	**Cause of death:** Acute yew intoxication **Manner of death:** Suicide	Our case II 2022
**39-year-old male**, found dead lying in a meadow under the forest. Near the body, a plastic bag containing *T. baccata* needles. Medical history: psychiatric illness 2 months before his death, he had already consumed yew.	Green needles of *T. baccata* plant in oesophagus and stomach contents Marked swelling of the lungs Brain swelling Petechiae in the soft coverings of the skull Mydriasis	Stomach content and urine: 3,5- dimetoxyphenol	GC-MS, LC-MS Q-TOF	**Cause of death:** heart failure due to *T. baccata* intoxication **Manner of death:** Suicide	Our case III

## Data Availability

Not applicable.

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
