# Peer review of "Natural Taxanes: From Plant Composition to Human Pharmacology and Toxicity"

_ijms, 2022, doi:10.3390/ijms232415619_

Round 1

Reviewer 1 Report

1. What is the main question addressed by the research?   Ans: The authors explain the importance of taxanes and its pharmacological activity.
2. Do you consider the topic original or relevant in the field? Does it
address a specific gap in the field?   Ans: The authors cover almost all pharmacological activity of taxanes. The best part is it covers toxicity study.
3. What does it add to the subject area compared with other published
material? Ans: Similar article is published Škubník J, Pavlíčková V, Ruml T, Rimpelová S. Current Perspectives on Taxanes: Focus on Their Bioactivity, Delivery and Combination Therapy. Plants. 2021; 10(3):569. https://doi.org/10.3390/plants10030569 so authors should add how this article differ form previous published paper.

4. What specific improvements should the authors consider regarding the
methodology? What further controls should be considered?   Ans: In introduction part, the importance of this article.
5. Are the conclusions consistent with the evidence and arguments presented
and do they address the main question posed? Ans: Yes   6. Are the references appropriate?   Ans: Yes
7. Please include any additional comments on the tables and figures.   Ans: Tables are fine

Author Response

Reviewer 1

English language and style

( ) English very difficult to understand/incomprehensible
( ) Extensive editing of English language and style required
( ) Moderate English changes required
( ) English language and style are fine/minor spell check required
(x) I don't feel qualified to judge about the English language and style

Comments and Suggestions for Authors

What does it add to the subject area compared with other published
material?

Ans: Similar article is published Škubník J, Pavlíčková V, Ruml T, Rimpelová S. Current Perspectives on Taxanes: Focus on Their Bioactivity, Delivery and Combination Therapy. Plants. 2021; 10(3):569. https://doi.org/10.3390/plants10030569 so authors should add how this article differ form previous published paper. 

Authors answer: At first glance, the article you mentioned appears similar, especially in describing the mechanism of action of paclitaxel (PTX). We decided to briefly describe this mechanism of action as well, as it is essential for understanding the toxic effect of the taxanes themselves. The differences are as follows. Regarding the different formulations of PTX, we describe their pharmacokinetic parameters in humans and Škubník et al. describe their legal status in clinical trials and their combinations in therapy, composition, advantages and disadvantages of use. Furthermore, Škubník et al. describe the isolation of Taxol or optimization of its isolation, increasing the isolation yield, while we describe all the major constituents of Taxus spp and the methods by which this composition was ascertained. Moreover, in contrast to Škubník et al., we pay almost no attention at all to the biosynthesis of taxanes, as well as to their synthesis and semisynthesis, and we deal extensively with unwanted effects and their toxicity. We see the main differences between the two papers mainly in the above-mentioned chapters.

What specific improvements should the authors consider regarding the
methodology? What further controls should be considered?  

Ans: In introduction part, the importance of this article.

Authors answer: 

The importance of this article can also be seen in the last sections, where the adverse effects of PTX treatment on organ systems, mechanisms of toxicity and intoxication are summarized not only in the case of PTX, but also in the case of the use of the plant itself, often ending in death. We also point out that in addition to the observed unwanted toxic effects during treatment, their correlation with actual blood levels of taxane would be useful, especially in cases of patients non-responding to chemotherapy. It is similar to clinical cases of poisoning, where current levels of taxanes in the blood and their correlation with symptoms of acute intoxication are required.

This was added to the end of the Introduction section and highlighted in yellow.

Reviewer 2 Report

This is a very interesting and comprehensive review on taxanes with a focus on the phytochemical, pharamcological and toxicological aspects. I read the paper with considerable interest and believe that it will also be of interest to many members of the community. The only point where I personally see some room for improvement would be to compile more information into tables, e.g. section 2.1: much of the information could be listed in a table, rather than describing it in flow text. This would make reading this otherwise fine and interesting review much easier.

Author Response

Reviewer 2

English language and style

( ) English very difficult to understand/incomprehensible
( ) Extensive editing of English language and style required
( ) Moderate English changes required
(x) English language and style are fine/minor spell check required
( ) I don't feel qualified to judge about the English language and style

Authors answer:

The entire manuscript was checked and revised for English language

This is a very interesting and comprehensive review on taxanes with a focus on the phytochemical, pharamcological and toxicological aspects. I read the paper with considerable interest and believe that it will also be of interest to many members of the community. The only point where I personally see some room for improvement would be to compile more information into tables, e.g. section 2.1: much of the information could be listed in a table, rather than describing it in flow text. This would make reading this otherwise fine and interesting review much easier.

Authors answer: Thank you for the suggestion. However, if possible, we would leave section 2.1 unchanged. In our opinion, the information content from Citations 24 and 25 at the beginning of the chapter is too limited to be placed in a next table. Nevertheless, the twelfth line of this chapter already begins with the summary from the extensive Table 2, where we wanted to highlight the most interesting information resulting from the mentioned table. To make another table from the summarised information arising from Table 2 seems confusing even for us.
